# FastAvatar: Rapid 3D Gaussian Splatting Face Avatar Generation from a Single Image

## Abstract

This paper presents a method to infer a 3D face avatar model from a single arbitrarily posed image, using the 3D Gaussian Splatting (3DGS) framework. Inference of a full 3DGS face model from one image is a highly ill-posed problem, requiring the estimation of hundreds of thousands, often well over a million, per-Gaussian appearance and structural parameters. To address this challenge, we draw inspiration from the classical morphable face models literature, in which individual identities are well-described as compact deformations (residuals) with respect to a canonical template face model, thereby easing the learning task. We propose leveraging such a template-plus-residuals strategy, but in the unstructured 3DGS parameter space. Rather than predicting absolute 3DGS parameters from scratch given an input face image, our proposed algorithm, FastAvatar, learns to map a face image to residual parameter values with respect to a canonical 3DGS template learned over prior multi-view face data. We couple the feed-forward prediction with a rapid inference-time latent refinement to maximize appearance fidelity to the observed image. Our evaluations on the Nersemble benchmark demonstrate that FastAvatar can generate 3DGS face models (∼600K parameters) in approximately 3 seconds, with state-of-the-art reconstruction accuracy (24.01 dB PSNR and 0.91 SSIM) compared to existing feed-forward, optimization, and diffusion baselines. Our work demonstrates that residual learning offers a tractable and high-fidelity approach to image synthesis in the popular 3DGS framework.

## 1 Introduction

Creating 3D face models from images is a long-standing problem in computer vision and graphics, and is of significant current interest in digital avatar applications such as virtual reality, gaming, and content creation. 3D face model frameworks that enable fast and high-resolution novel view rendering performance from one or few input views are needed to support these applications. Classical parametric face models based on simple statistical approaches (Blanz & Vetter, 2003; 2023; Li et al., 2017a) offer real-time fitting and rendering speed, but are limited in their expressive power. In contrast, recent approaches based on the Neural Radiance Field (NeRF) (Mildenhall et al., 2021) and 3D Gaussian Splatting (3DGS) (Kerbl et al., 2023) neural rendering frameworks offer state-of-the-art expressive power, with 3DGS even enabling real-time rendering speed. However, these algorithms have two critical limitations. First, they require multi-view captures of the subject, which is only possible in a lab setup. Second, they involve expensive per-subject optimizations using these multi-view captures with fitting times ranging from minutes to hours, restricting their deployment applications.

In this study, we aim to move beyond purely optimization-based avatar construction and propose a fast, robust framework to reconstruct a high-quality 3DGS face model from a single input image. Purely feed-forward face generation approaches, typically based on GANs (Goodfellow et al., 2014) or diffusion models (Ho et al., 2020), are often constrained to (near-)frontal viewpoints due to data bias and struggle to generalize under large pose variation, frequently exhibiting artifacts or identity drift when viewed from novel angles (Gerogiannis et al., 2025; Chan et al., 2022). At the same time, fully optimization-driven 3DGS reconstruction methods deliver high fidelity but require minutes of per-subject fitting and are unsuitable for interactive user applications. This motivates the need for a single-view reconstruction framework that combines the speed

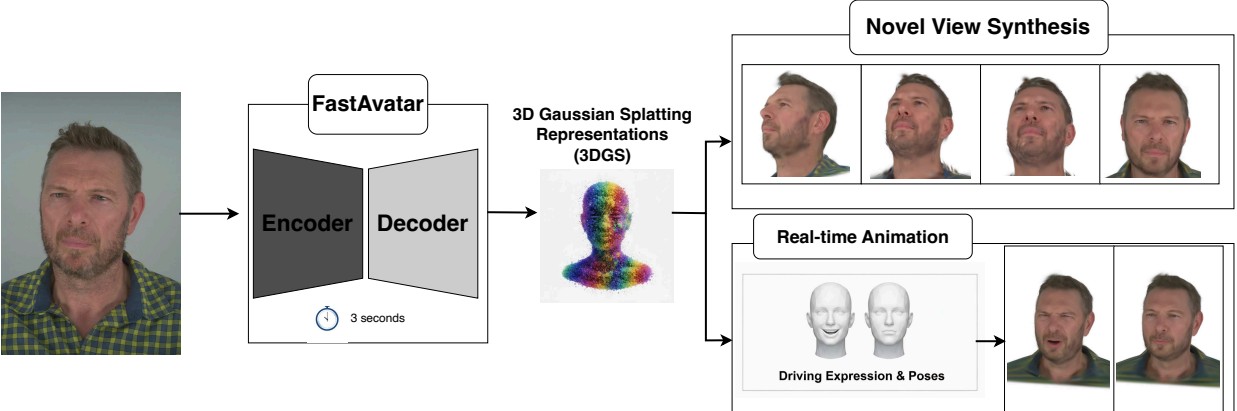

Figure 1: **FastAvatar: Amortized prediction of high-dimensional 3D Gaussian parameters from a single image.** Given an arbitrary-pose 2D observation, our feed-forward framework instantly maps the input to over 600K continuous 3DGS parameters. By leveraging a canonical geometric prior and a lightweight ∼3-second refinement, FastAvatar bypasses the instability of traditional per-subject optimization to yield a geometrically coherent representation. The predicted structured output is highly robust, supporting real-time novel-view synthesis and FLAME-driven expression animation across all viewpoints.

and stability of direct prediction with the fidelity of optimization, while remaining robust to large input pose variation and maintaining faithful identity preservation. We emphasize that, although our training pipeline uses multi-view data to build the canonical template and train the encoder-decoder network, the trained system needs only a single image at inference time. We use the multi-view data solely to build the geometric prior offline, and we need no multi-view capture to reconstruct a new subject.

To address these challenges, we present FastAvatar, a fast framework for 3DGS face reconstruction from a single image with arbitrary pose. FastAvatar is driven by two key insights. First, inspired by classical morphable face models (Blanz & Vetter, 2003), we construct a canonical 3DGS "template" by averaging Gaussian parameters across a set of subject-specific models. Compared to prior work that relies on random initialization or joint template optimization (Saunders et al., 2025; Qian et al., 2024a; Yan et al., 2025; Liu et al., 2024), this simple averaging procedure yields a compact and highly stable geometric prior. Consequently, at inference time, FastAvatar can efficiently reconstruct a new identity using a 3DGS Residual Prediction Network to infer only small, subject-specific residuals from the template.

Second, to achieve identity preservation even under extreme viewpoints, we constrain the encoder to map all views of the same individual to a shared latent vector using pre-trained face recognition features (Deng et al., 2019) and pose-invariant regression. The decoder then maps this identity embedding to Gaussian parameter residuals that refine the template into a subject-specific 3D representation. As a result, FastAvatar produces stable, 3D-consistent geometry even from extreme input poses, avoiding the view-dependent artifacts or identity drift commonly observed in unconstrained image-to-3D prediction networks. We subsequently refine the feed-forward predicted model with a lightweight inference-time optimization routine to recover high-quality details.

We evaluated FastAvatar with several quantitative and qualitative experiments using the Nersemble test dataset (Kirschstein et al., 2023) and compared against recent 3DGS-based and diffusion-based avatar methods. Given three distinct input poses (frontal and two extreme views), we measured reconstruction quality across 15 novel viewpoints. Results show that FastAvatar achieves 24 dB PSNR in approximately 3 seconds, outperforming feed-forward baselines such as GAGAvatar (Chu & Harada, 2024) (16 dB), LAM (He et al., 2025)(14 dB), and running 600× faster than purely optimization-driven approaches such as GaussianAvatars (Qian et al., 2024a), FlashAvatar (Xiang et al., 2024), DiffusionRig (Ding et al., 2023), and Arc2Avatar (Gerogiannis et al., 2025). FastAvatar maintains stable identity and reconstruction quality across large input poses, from frontal to extreme profiles, whereas existing feed-forward or template-free

approaches tend to degrade or drift. We also provide qualitative examples using FastAvatar to perform FLAME-guided expression animation from a single image. By combining high fidelity, robustness to pose, and rapid reconstruction, FastAvatar broadens the applicability of 3DGS to practical and interactive avatar creation.

In summary, our key contributions are:

- **Canonical Residual Formulation:** We formulate high-dimensional 3DGS prediction as a residual regression task. Anchoring predictions to an averaged, topologically consistent 3D face template significantly constrains the non-convex search space, accelerating convergence and preventing structural artifacts.

- **Empirical Validation and Real-Time Latent Control:** Our framework achieves a strong single-image 3D face reconstruction performance (24.01 dB PSNR). Furthermore, our structured latent space uniquely enables real-time attribute control via latent traversal, allowing smooth, zero-shot appearance editing without costly re-optimization.

## 2 Related Work

### 2.1 Parametric 3D Face Models

Recovering a high-quality 3D face from images is a long-standing problem in computer vision and graphics. The seminal 3D Morphable Model (3DMM) (Blanz & Vetter, 2003; Smith et al., 2020; Blanz & Vetter, 2023; Li et al., 2020) represents facial geometry and appearance using statistical mesh-based bases learned from many subjects. FLAME (Li et al., 2017a) extends these models to capture expression variations and articulated jaw and neck motion using thousands of 3D head scans (Feng et al., 2021; Daněček et al., 2022). Both 3DMM- and FLAME-based methods estimate parameters via inverse rendering or neural regression, but their low-rank bases limit expressiveness for high-frequency facial details such as subtle skin variation and hair patterns.

### 2.2 Optimization-Based Neural Rendering

In the past five years, the field of neural rendering has opened new capabilities in 3D face reconstruction. Neural Radiance Fields (NeRF) (Mildenhall et al., 2021) and their dynamic variants (Pumarola et al., 2021; Park et al., 2021b; Li et al., 2021; Fang et al., 2022) enable photorealistic view synthesis by representing scenes as continuous volumetric fields. NeRFs have also been developed for face reconstruction (Tretschk et al., 2021; Park et al., 2021a; Zhuang et al., 2022; Hong et al., 2022; Gafni et al., 2021; Buehler et al., 2024; Trevithick et al., 2023). However, NeRFs are typically slow to train and render, often requiring hours to fit a new scene.

3D Gaussian Splatting (3DGS) is a compelling alternative to NeRF that explicitly represents scenes as a set of anisotropic Gaussian primitives with learned parameters (Kerbl et al., 2023). 3DGS supports high-quality rendering at real-time frame rates and has been adapted to face and head modeling, achieving impressive visual quality (Xu et al., 2024b; Dhamo et al., 2024; Wei et al., 2025; Ma et al., 2024c). Recent studies have started to combine parametric meshes with 3DGS to gain the advantages of dense correspondence and identity-expression disentanglement from meshes, along with efficient, high-quality rendering from 3DGS (Qian et al., 2024a; Shao et al., 2024; Xiang et al., 2024; Kocabas et al., 2024; Xu et al., 2024a; Qian et al., 2024b; Saunders et al., 2025; Yan et al., 2025; Hu et al., 2024; Wang et al., 2025a; Giebenhain et al., 2024).

These methods typically attach Gaussians to vertices (or surface patches) of a fitted parametric head mesh and optimize per-Gaussian parameters (means, scales, appearances) to the observed image(s). In addition, while most of these methods require multi-view capture, several have been extended to the far more under-constrained single-view setting (Ki et al., 2024; Li et al., 2023b; Ma et al., 2023; Tran et al., 2024; Yang et al., 2020; Zielonka et al., 2022; Li et al., 2024; He et al., 2025). While these hybrid 3DGS approaches can

produce detailed reconstructions and can fit scenes far faster than NeRFs (often nearly $10\times$ faster), their fitting times are still far from real-time due to their reliance on per-face iterative optimization.

Unlike prior mesh-anchored 3DGS methods that initialize templates randomly or jointly optimize them with the decoder (entangling with the latent space), we fix the template by averaging subject-specific optimized 3DGS models and only learn offsets—yielding smaller, better-conditioned residuals, removing the $(K \times P)$ parameter block, and improving stability and generalization. Here, $K$ denotes the number of Gaussians (we use $K = 10{,}144$ in our experiments) and $P = 59$ denotes the number of parameters per Gaussian, which we break down as 3 (position) $+ 3$ (scale) $+ 4$ (rotation quaternion) $+ 1$ (opacity) $+ 48$ (spherical harmonic coefficients).

### 2.3 Feed-Forward and Generative Face Reconstruction

In contrast to iterative optimization approaches, feed-forward methods directly map input images to outputs, either 3D models or novel views, resulting in real-time fitting speed. Several feed-forward methods directly predict parameters to NeRF or 3DGS models given one or more input images (Zheng et al., 2023; Li et al., 2023b;a; Chu & Harada, 2024; Ma et al., 2024a; Yang et al., 2024; Chu et al., 2024; Ye et al., 2024; Wang et al., 2025b). These approaches work well on frontal or near-frontal inputs with limited pose variation, but often exhibit pose-dependent inconsistencies; reconstructions from non-frontal inputs can differ noticeably in identity and geometry compared to frontal inputs.

Another family of feed-forward methods use generative models such as diffusion models (Paraperas Papantoniou et al., 2024; Gerogiannis et al., 2025; Shiohara & Yamasaki, 2024; Chen et al., 2024; Ma et al., 2024b; Shi et al., 2024; Wei et al., 2023; Xiao et al., 2025; Taubner et al., 2025) or GANs (Chan et al., 2022; Sun et al., 2023; Zhao et al., 2024; Deng et al., 2024; Karras et al., 2019; 2020; An et al., 2023; Gecer et al., 2020; Lattas et al., 2023) to directly infer novel views by exploiting data-driven priors. Generative models now offer photorealistic synthesis quality, but do not typically construct explicit 3D geometry, and thus often produce distortions, drifting facial structures, and identity hallucinations with pose changes.

## 3 Method

Given a single face image $I$, our goal is to reconstruct a complete 3DGS model that enables high-quality novel-view synthesis and animation. The main challenge is to infer hundreds of thousands of Gaussian parameters from a single 2D observation under extreme pose variation. FastAvatar addresses this challenge through a two-stage framework that leverages a canonical 3DGS template and prior data to perform pose-invariant residual prediction (see Fig. 2). We first build a base template ($\mathcal{T}$) by averaging Gaussian (structural and appearance) parameters across subject-specific 3DGS models, with all Gaussians placed at anatomically consistent surface locations using a tracked parametric face model (Li et al., 2017a). This provides a strong geometric basis, ensuring consistent correspondence of parameters across subjects. In the first stage (feed-forward inference), FastAvatar uses a 3DGS Residual Prediction Network to map the input image $I$ to a pose-independent identity embedding $\omega$, and decodes this embedding into per-Gaussian residual parameters ($\Delta G$) with respect to the template. This yields a structurally robust coarse 3DGS model in real time. In the second stage (inference-time refinement), we achieve high-fidelity rendering by optimizing this coarse model against the input image. Crucially, rather than performing unconstrained optimization on the raw Gaussian parameters, we backpropagate the reconstruction loss to fine-tune only the latent identity code $\omega$ and the decoder network. This targeted refinement recovers intricate personalized details while preserving the learned geometric prior. We reiterate that we use multi-view data only during offline training, where it fits the per-subject 3DGS models that form the template and supervises the encoder-decoder network. At inference time, both stages use a single input image, and a new subject needs no multi-view capture. We next describe the 3DGS formulation and details of FastAvatar.

### 3.1 Preliminaries: 3D Gaussian Splatting (3DGS)

A 3D Gaussian Splatting (3DGS) model (Kerbl et al., 2023) represents a scene using $K$ anisotropic Gaussians $\mathcal{M} = \{G_k\}_{k=1}^{K}$, each defined by geometric and appearance parameters: center $\mu_k \in \mathbb{R}^3$, opacity $\alpha_k \in [0,1]$,

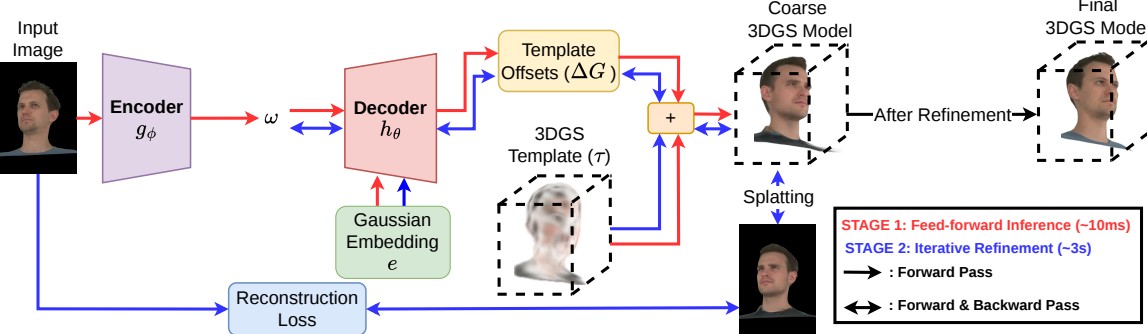

Figure 2: **FastAvatar inference pipeline.** Our framework reconstructs a high-fidelity 3DGS avatar from a single image using a two-stage process. **Stage 1 (Red arrows):** In a fast feed-forward pass (∼10ms), the encoder $g_\phi$ extracts a pose-invariant identity embedding $\omega$. The decoder $h_\theta$ maps $\omega$ to per-Gaussian offsets $\Delta G$, which are added to the canonical template $\mathcal{T}$ to yield a structurally robust coarse 3DGS model. **Stage 2 (Blue arrows):** A targeted inference-time refinement (∼3s) optimizes the coarse model against the input image. Crucially, the reconstruction loss backpropagates only to update the latent code $\omega$ and decoder $h_\theta$, bypassing raw 3DGS parameter optimization to preserve geometric stability while recovering high-frequency personalized details.

spherical harmonic (SH) color coefficients $c_k$, and a covariance matrix $\Sigma_k = R_k S_k^2 R_k^T$ factored into a rotation $R_k$ (quaternion $q_k$) and diagonal scale $S_k$. Each Gaussian has 59 parameters including geometry (center, scale, rotation), opacity, and 48 SH appearance coefficients. The framework uses differentiable rasterization, in which Gaussians are alpha-blended along each ray: $C = \sum_{d=1}^{D} c_d \alpha_d \prod_{j<d}(1 - \alpha_j)$, with $c_d$ evaluated from SH coefficients and $\alpha_d$ incorporating opacity and projected density. Here, $d$ indexes the $D$ Gaussians intersected along a camera ray, sorted by depth; $c_d$ is the color evaluated from the spherical harmonic coefficients of Gaussian $d$; and $\alpha_d$ is the effective opacity of Gaussian $d$, incorporating its learned opacity and its projected 2D density at the pixel location.

## 3.2 Template 3DGS Face Model Construction

Inspired by classical morphable models (Blanz & Vetter, 2023), we first construct a data-driven template 3DGS model $\mathcal{T}$.

In order to ensure a common structure (and have per-Gaussian correspondences) across 3DGS models of all subjects, we follow the strategy of GaussianAvatars (Qian et al., 2024a) and place Gaussians using the standard FLAME mesh representation of each face (Li et al., 2017a). We place one Gaussian at the center of each triangular face of the FLAME mesh and add additional Gaussians for the upper and lower teeth. This produces a canonical set of $K$ mesh-attached Gaussians that share consistent semantic meaning across subjects and provide a stable geometric reference for learning and animation.

As shown in Fig.3, we construct template $\mathcal{T}$ by first optimizing individual 3DGS models $\{\mathcal{M}_i\}_{i=1}^N$ for each training subject, with Gaussians initialized at FLAME-aligned locations. We then apply standard 3DGS optimization (Ye et al., 2025) to recover subject-specific geometries and appearances.

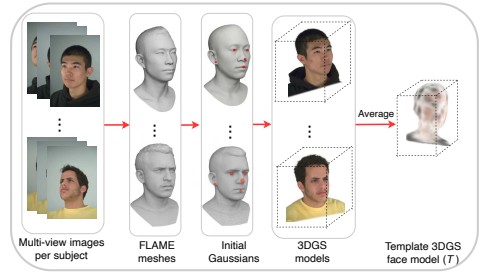

Figure 3: **Template 3DGS face model construction.** FastAvatar constructs a template 3DGS face model $\mathcal{T}$ by averaging parameters of Gaussians across 3DGS models fit on a training set of subjects.

Finally, we average the parameters of corresponding Gaussians across subjects to obtain $\mathcal{T}$. In our experiments, $\mathcal{T}$ consists of 10,144 Gaussians, yielding 598,496 parameters in total.

### 3.3 3DGS Residual Prediction Network

FastAvatar's 3DGS Residual Prediction Network maps an input image $I$ to residual Gaussian parameters that deform the template $\mathcal{T}$ into a subject-specific 3DGS model. To achieve this, we structure the network as an encoder-decoder architecture. Because $I$ can appear under arbitrary viewpoints, we design the encoder $g_\phi$ to produce a *pose-invariant* latent code that captures only identity-specific information. Subsequently, the decoder $h_\theta$ projects this latent embedding into the spatial and appearance offsets ($\Delta$) applied to $\mathcal{T}$.

We found that training the encoder-decoder in an end-to-end manner leads to a degenerate solution in which the network always predicts an average face. To avoid this, we adopt a decoupled training strategy. We first pretrain the decoder while treating the latent codes for all training subjects as learnable variables. This initial phase forces the decoder to learn a smooth and generalizable latent space for predicting Gaussian residuals, rather than memorizing per-identity solutions. Once the decoder is trained, we freeze it and optimize the encoder to map images into this learned latent space. We guide the encoder with features from a large-scale face recognition model to ensure pose invariance and strong identity discrimination. This decoupled approach yields an encoder that generalizes reliably to unseen identities and expressions, while the decoder provides stable geometry prediction anchored to the canonical template.

#### 3.3.1 Decoder Design and Optimization

The decoder $h_\theta$ is a shallow MLP that maps a pose-invariant identity code $w \in \mathbb{R}^{|w|}$ (with $|w| = 512$) and a Gaussian embedding $e_k \in \mathbb{R}^{|e|}$ (with $|e| = 32$) to residual parameters $\Delta G_k = \{\Delta\mu_k, \Delta s_k, \Delta q_k, \Delta\alpha_k, \Delta c_k\}$ for Gaussian $k$, whose components have dimensionalities $\Delta\mu_k \in \mathbb{R}^3$ (position), $\Delta s_k \in \mathbb{R}^3$ (scale), $\Delta q_k \in \mathbb{R}^4$ (rotation quaternion), $\Delta\alpha_k \in \mathbb{R}^1$ (opacity), and $\Delta c_k \in \mathbb{R}^{48}$ (spherical harmonic coefficients). The identity code $w$ captures subject-dependent properties, while $e_k$ provides localized context for each Gaussian.

We then apply the residuals to the template Gaussian parameters(means, scale, rotation, opacity, SH coefficients). Specifically, we get the final parameters by: $\mu_k = \mu^{\mathcal{T}} + \Delta\mu_k$, $s_k = s_k^{\mathcal{T}} \cdot \exp(\Delta s_k)$, $q_k = \text{normalize}(q_k^{\mathcal{T}} + \Delta q_k)$, $\alpha_k = \sigma(\text{logit}(\alpha_k^{\mathcal{T}}) + \Delta\alpha_k)$, and $c_k = c_k^{\mathcal{T}} + \Delta c_k$.

We implement the decoder with a shallow MLP. We initialize Gaussian embeddings $\{e_k\}$ using sinusoidal positional encodings of their canonical FLAME coordinates, and identity codes $\{w_i\}_{i=1}^N$ from a standard Normal distribution. FLAME (Li et al., 2017a) is a parametric 3D head model that represents facial geometry as a triangular mesh with fixed topology (9,976 faces). The *canonical FLAME coordinates* are the 3D $(x, y, z)$ positions of these triangle centers (where we place the Gaussians) when the mesh is in its rest state, i.e., neutral expression and zero pose. Because the topology is fixed, the $k$-th triangle always corresponds to the same facial region across all subjects, so the sinusoidal encodings of these coordinates give each Gaussian a stable, semantically consistent embedding $e_k$. During decoder pretraining, we jointly optimize $h_\theta$, $\{w_i\}$, and $\{e_k\}$ using

$$\mathcal{L}_{\text{dec}} = \lambda_1 \mathcal{L}_{\text{LPIPS}} + \lambda_2 \mathcal{L}_1 + \lambda_3 \mathcal{L}_{\text{SSIM}} + \lambda_4 \mathcal{L}_2(\Delta\mu) + \lambda_5 \mathcal{L}_2(\Delta s), \tag{1}$$

computed between rendered predictions and ground-truth images. LPIPS, SSIM, and $L_1$ promote perceptual and photometric accuracy, while the $L_2$ regularizers keep residuals small and stable. We set the loss weights to $\lambda_1 = 0.1$, $\lambda_2 = 0.6$, $\lambda_3 = 0.3$, and $\lambda_4 = \lambda_5 = 5 \times 10^{-3}$.

#### 3.3.2 Encoder Design and Optimization

The encoder $g_\phi$ predicts a pose-invariant latent code for the face depicted in $I$, from which the decoder can infer a full 3DGS representation. We construct the encoder as a composition $g_\phi = g_{\text{MLP}} \circ g_{\text{FR}}$, where $g_{\text{FR}}$ is a pretrained face recognition backbone and $g_{\text{MLP}}$ is a lightweight projection network. The face recognition model provides identity features that are invariant to viewpoint, while the projection MLP maps these features to the latent space used by the decoder.

Given the precomputed identity codes $\{w_i\}_{i=1}^N$ learned during decoder training, we train $g_\phi$ to map all images of the same subject to their code. The encoder loss is

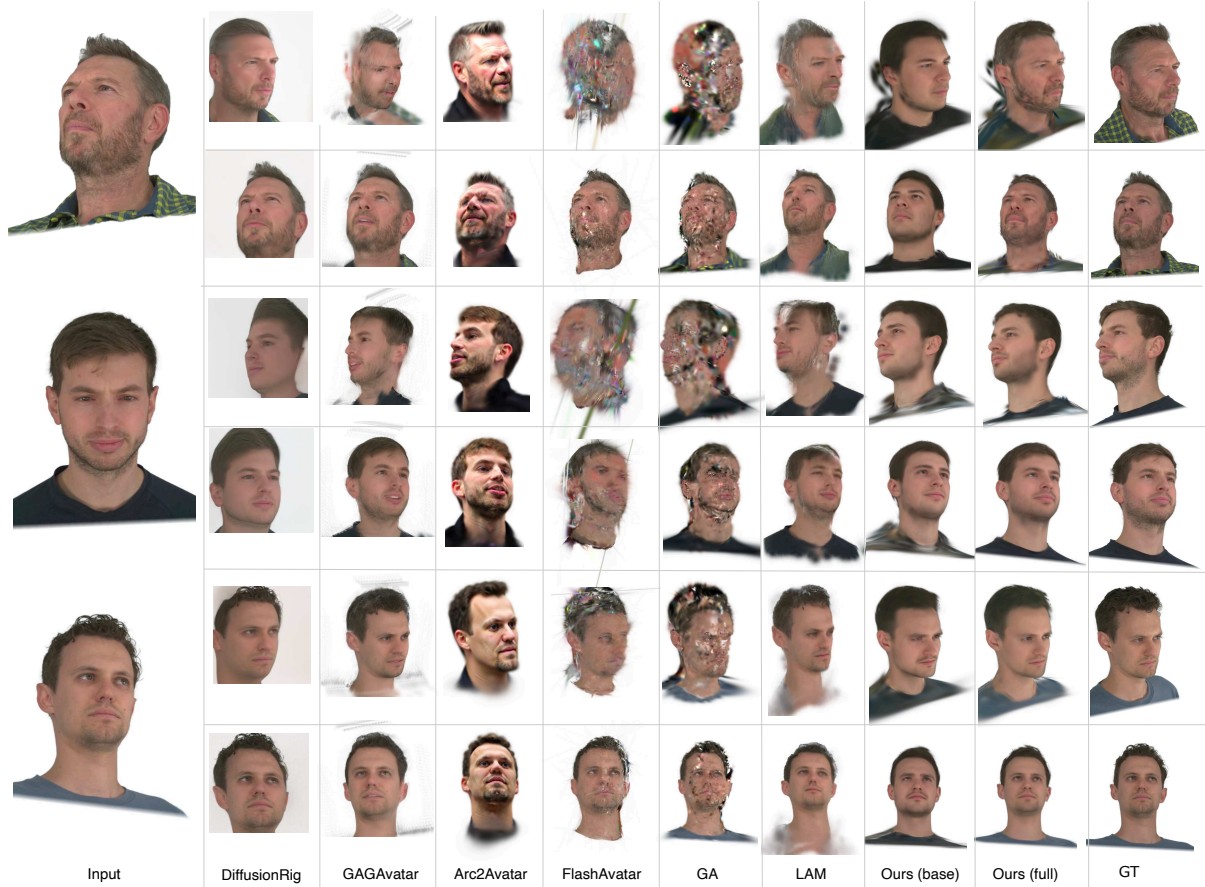

| Input | DiffusionRig | GAGAvatar | Arc2Avatar | FlashAvatar | GA | LAM | Ours (base) | Ours (full) | GT |

Figure 4: **Qualitative comparison on single-image novel-view synthesis.** Given a single arbitrary-view input (left), we compare FastAvatar (base and full) with DiffusionRig (Ding et al., 2023), GAGA-vatar (Chu & Harada, 2024), LAM (He et al., 2025), Arc2Avatar (Gerogiannis et al., 2025), FlashAvatar (Xiang et al., 2024), and GaussianAvatars (GA) (Qian et al., 2024a). GAGAvatar and Arc2Avatar operate in their own canonical spaces; following prior work, we align their outputs to our coordinate frame via PnP (details in Supplementary), though small residual shifts may remain. Diffusion-based and feed-forward baselines struggle under large input poses, often producing blurry textures, synthetic-looking faces, or distorted geometry. GA and FlashAvatar, which require multi-view fitting, degrade noticeably when extended to the single-view setting. In contrast, FastAvatar maintains coherent geometry and identity across wide viewpoint changes; the full model further sharpens appearance. Additional examples, including more poses, expressions, and identity-similarity metrics, are provided in the Supplementary.

$$\mathcal{L}_{\mathrm{enc}}(w, \hat{w}) = \mathcal{L}_2(w, \hat{w}) + \lambda_{\mathrm{cos}}\, \mathcal{L}_{\mathrm{cos}}(w, \hat{w}), \tag{2}$$

where $\hat{w} = g_\phi(I)$ is the predicted embedding and $\mathcal{L}_{\mathrm{cos}}$ is the cosine distance, which we weight by $\lambda_{\mathrm{cos}} = 0.1$. This training strategy encourages the encoder to generalize to unseen identities and arbitrary poses.

### 3.4 Two-Stage Inference Pipeline

Given an input image $I$ from an arbitrary viewpoint, our two-stage inference pipeline begins with a single feed-forward pass (Stage 1) that predicts the parameter residuals needed to deform the canonical template. This establishes a stable, 3D-consistent coarse geometry anchored by our learned prior. To capture high-

fidelity personalized details, we proceed to Stage 2: a targeted inference-time refinement. For approximately 300 iterations ($\sim 3$ seconds on an A100 GPU), we optimize the latent identity code $\omega$ and the decoder weights $h_\theta$ using the same reconstruction loss as Eq.(1). Crucially, unlike single-view methods that directly optimize raw Gaussian attributes (Saunders et al., 2025; He et al., 2025), a practice that frequently degrades underlying geometry and compromises multi-view consistency, our two-stage approach strictly prohibits direct updates to the 3DGS parameters. Instead, all appearance adjustments must flow through the network's learned latent space. This constraint ensures that the refined avatar perfectly retains its canonical structural integrity without introducing view-specific artifacts. Once refined, the final subject-specific Gaussians can be animated and rendered at real-time frame rates.

## 3.5 Animation and Reenactment

Because our predicted 3DGS parameters are directly tied to a topologically consistent template, FastAvatar naturally supports facial animation. To achieve this, our animation module takes advantage of the explicit mesh-anchored structure established during template construction (Sec. 3.2), akin to GaussianAvatars (Qian et al., 2024a). Since the Gaussians are already bound to specific structural locations (including supplemental teeth anchors), they simply inherit the fixed skinning weights $\{w_{k,j}\}_{j=1}^J$ from the underlying FLAME mesh, allowing the reconstructed avatar to be driven by novel expression parameters.

Given target FLAME expression or pose parameters, we obtain the joint transformations $\{A_j\}_{j=1}^J$ and animate each Gaussian using standard linear blend skinning (LBS). For Gaussian $k$, the local canonical parameters $(\mu'_k, \sigma'_k, \mathbf{r}'_k)$ are mapped to global space via

$$(\mu_k^{\text{anim}}, \sigma_k^{\text{anim}}, \mathbf{r}_k^{\text{anim}}) = \mathcal{T}_{\text{local}\rightarrow\text{global}}\big((\mu'_k, \sigma'_k, \mathbf{r}'_k), \, T_k\big),$$

where $\mathcal{T}_{\text{local}\rightarrow\text{global}}$ denotes the standard SE(3)-based Gaussian frame transform, and $T_k = \sum_{j=1}^J w_{k,j} A_j$ is the blended transformation for Gaussian $k$. Here $w_{k,j}$ is the fixed linear blend skinning weight of Gaussian $k$ with respect to FLAME joint $j$ (inherited from the underlying mesh), and $A_j$ is the transformation matrix of joint $j$ induced by the target FLAME pose and expression. This updates position, rotation, and scale consistently with the FLAME-driven deformation.

This procedure yields smooth expression and pose changes without additional optimization, enabling real-time reenactment after 3DGS model reconstruction. In addition, due to the well-structured latent space, our framework also supports real-time animation to attributes beyond expression (see Fig. 9 and Sec.4.5).

## 4 Experiments & Results

We evaluated FastAvatar on the task of single-view 3D face reconstruction using the NeRSemble dataset (Kirschstein et al., 2023). This dataset features 422 subject identities performing various facial expressions, captured simultaneously from 16 calibrated viewpoints ranging from frontal to extreme profile poses. We partitioned the dataset into 410 identities for training and 12 held-out identities for evaluation.

**Baselines.** We compare FastAvatar against recent state-of-the-art 3DGS and neural avatar methods: (1) *Optimization-driven*: GaussianAvatars (Qian et al., 2024a), FlashAvatar (Xiang et al., 2024); (2) *Feed-forward*: GAGAvatar (Chu & Harada, 2024), LAM (He et al., 2025); (3) *Diffusion-based*: Arc2Avatar (Gerogiannis et al., 2025), DiffusionRig (Ding et al., 2023). We use the official implementations and recommended configurations for all methods. Because Arc2Avatar, GAGAvatar, and LAM operate in their own canonical spaces, we estimate camera intrinsics and extrinsics by PnP-aligning their reconstructed geometry to our canonical template (see Supplementary). Throughout the experiments, Ours(base) denotes the feed-forward encoder–decoder prediction after Stage 1 (geometry only), and Ours(full) includes the subsequent appearance refinement, Stage 2.

**Metrics.** We evaluate reconstruction accuracy using PSNR, SSIM (Wang et al., 2004), LPIPS (Zhang et al., 2018), and Identity Similarity (Deng et al., 2019). For each test identity, we use one of the 16 views as the input and evaluate rendering quality on the remaining 15 views, repeating this for all input viewpoints. All runtime numbers are measured on a single NVIDIA A100 (40GB).

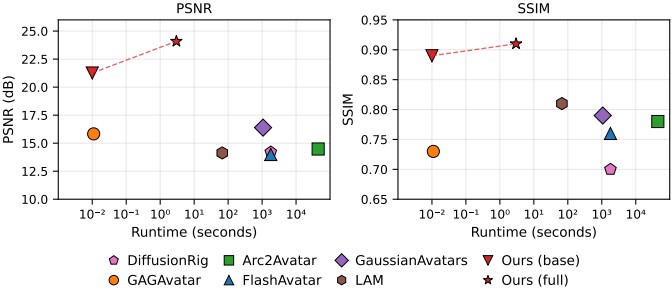 

Figure 5: **Reconstruction quality vs. runtime.** Our feed-forward base model already produces strong reconstructions (21.17 dB PSNR, 0.89 SSIM). With only ~3 seconds of refinement, our full method reaches state-of-the-art quality (24.01 dB PSNR, 0.91 SSIM).

Figure 6: **Generalization to out-of-distribution identities.** Left column = single input image. The remaining columns = novel-view renderings from the reconstructed 3DGS model by FastAvatar.

**Implementation Details.** We constructed the template $\mathcal{T}$ from the 410 subject-specific 3DGS models optimized for 7,000 iterations using all 16 views, each trained on a random expression. We used VHAP (Qian, 2024) to extract FLAME parameters and camera poses. Following GaussianAvatars, we place one Gaussian at the center of each FLAME mesh face (9,976 faces) and add 168 Gaussians for the upper and lower teeth, resulting in $K = 10,144$ mesh-attached Gaussians with consistent semantic correspondence across all subjects. For training the main encoder-decoder network, we sampled 6 distinct expressions per training identity and assigned a unique latent code to each identity-expression pair, resulting in a total of 2,460 training samples. Please refer to Supplementary for additional implementation and training details.

## 4.1 Novel View Reconstruction Results

We first present qualitative comparisons in Fig. 4 for three sample test cases. GAGAvatar, LAM, and DiffusionRig exhibit strong degradation under non-frontal inputs, producing broken geometry, scattered points, or blurry outputs. Arc2Avatar often yields synthetic facial textures and may alter expressions (e.g., adding an open mouth). FlashAvatar and GaussianAvatars (GA), which rely on multi-view optimization, struggle in the single-view setting and produce noisy or incomplete reconstructions for challenging viewpoints. These two methods can be viewed as refinement-only analogs of our system: they optimize Gaussian parameters directly without a geometric prior, highlighting the role of FastAvatar's encoder–decoder stage in establishing stable, 3D-consistent structure from a single image.

In contrast, FastAvatar delivers stable geometry and consistent identity across all novel views. The feed-forward prediction (Ours(base)) already provides coherent 3D structure from a single input, while the refinement stage (Ours(full)) further enhances appearance, capturing subtle facial details, hair, and clothing with improved fidelity. Together, these stages achieve high-quality novel-view synthesis that remains accurate across large viewpoint changes.

Table 1 summarizes quantitative results for novel-view reconstruction. Ours (full) achieves the best performance across all metrics (24.01 dB PSNR, 0.91 SSIM, 0.19 LPIPS and 0.81 ID similarity), substantially outperforming all baselines. Ours (base) – the feed-forward encoder–decoder prediction – already provides strong geometry (21.17 dB PSNR, 0.89 SSIM) before the refinement phase.

## 4.2 Runtime Results

Fig. 5 presents the trade-off between reconstruction quality and fitting time for different methods. GAGAvatar and LAM operate in (milli-)seconds but produce far lower accuracy. Diffusion-based (DiffusionRig, Arc2Avatar) and optimization-driven (FlashAvatar, GaussianAvatars) methods require seconds-to-hours of per-subject optimization while still trailing our method in accuracy. Ours (full) achieves the best overall reconstruction with only ~3 seconds of refinement, representing a substantially better quality–speed trade-

Table 1: **Quantitative comparison on novel-view reconstruction.** FastAvatar achieves the best performance across all metrics. Methods marked with an asterisk (*) may have slightly underestimated scores due to minor residual misalignment after PnP-based canonical alignment.

| Method | PSNR ↑ | SSIM ↑ | LPIPS ↓ | ID Sim. ↑ |
|---|---|---|---|---|
| DiffusionRig | 14.21 | 0.70 | 0.29 | 0.65 |
| GAGAvatar* | 15.83 | 0.73 | 0.33 | 0.76 |
| Arc2Avatar* | 14.48 | 0.78 | 0.30 | 0.61 |
| FlashAvatar | 13.99 | 0.76 | 0.32 | 0.35 |
| GaussianAvatars | 16.39 | 0.79 | 0.30 | 0.39 |
| LAM | 14.13 | 0.81 | 0.34 | 0.77 |
| Ours (base) | 21.17 | 0.89 | 0.22 | 0.70 |
| Ours (full) | **24.01** | **0.91** | **0.19** | **0.81** |

Table 2: **Ablation results on FastAvatar design choices.** All results are reported for full version unless noted.

| Setting | PSNR ↑ | SSIM ↑ | Runtime (ms) |
|---|---|---|---|
| *Number of Gaussians K* (**Default:** 10,144) | | | |
| 5023 | 20.94 | 0.84 | 9 |
| 40,000 | 25.58 | 0.93 | 29 |
| *Loss Weights* (**Default:** Eq.1–2 | | | |
| w/o SSIM (dec.) | 22.03 | 0.88 | 10 |
| w/o CosSim (enc.) | 22.27 | 0.89 | 10 |
| *Gaussian Init.* (**Default:** Average Template) | | | |
| Random Init | 21.92 | 0.88 | 10 |
| Joint-Opt. | 22.84 | 0.89 | 10 |
| *Refinement Iterations* (**Default:** 300) | | | |
| 0 (feed-forward) | 21.17 | 0.89 | 10 |
| 600 | 24.32 | 0.91 | 6000 |
| **Ours (full)** | 24.01 | 0.91 | 3000 |

off than all prior approaches. Ours (base) runs in a single feed-forward pass and already reaches strong accuracy, highlighting the efficiency and stability of the encoder–decoder geometry prior.

### 4.3 Self- and Cross-Reenactment

Fig. 7 shows both self- and cross-reenactment results. In the self-reenactment examples, we drive the reconstructed subject using FLAME expression parameters extracted from other frames of the same identity. The outputs follow the target expressions while maintaining the subject's overall geometry and appearance. In the cross-reenactment examples, we use FLAME parameters from a different "driver" subject. The transferred expressions are reproduced on the source face without altering its identity, and the deformations remain stable across different expressions. These examples illustrate that the FLAME-guided deformation model allows FastAvatar to reproduce a range of expressions from a single reconstructed identity.

To probe temporal stability under dynamic motion, we further drive the avatar with a free-form LBS sequence that spans a continuous range of head poses and expressions (Fig. 7, bottom row). The avatar stays temporally stable under moderate pose and expression changes, while large movements can introduce stretching and tearing in the Gaussians, consistent with the known sensitivity of 3DGS to large non-rigid deformations. In addition, because FastAvatar reconstructs from a single image, the animation can only contain what the input reveals: if the input photo does not show teeth, the animated mouth will not contain them either.

### 4.4 Out-Of-Distribution Identities

We further test FastAvatar on identities outside the NeRSemble dataset. We obtain FLAME parameters for these images with EMOCA (Daněček et al., 2022) to align them with our canonical template. Fig. 6 shows that FastAvatar reconstructs identity-preserving 3DGS models from a single image of these out-of-distribution identities. We provide additional OOD subjects and viewpoints in the Supplementary.

We also test in-the-wild inputs sampled from FFHQ (Karras et al., 2019) and CelebA-HQ (Karras et al., 2018). For each image, we first run a face detector (Deng et al., 2020) to crop and mask out the face region, and then run FastAvatar on the masked input. Because these images have no ground-truth multi-view captures, we evaluate them qualitatively. Fig. 8 shows that FastAvatar recovers a reasonable avatar and identity-consistent novel views for many subjects, though some views contain hallucinations.

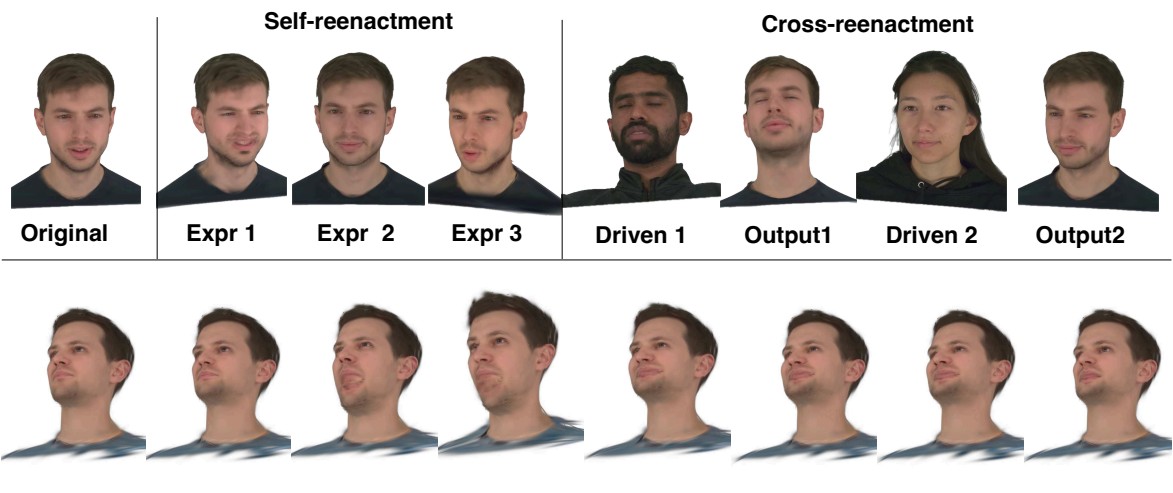

Figure 7: **Self- and cross-reenactment.** Starting from a single reconstructed face (left), FastAvatar can reproduce expressions from the same subject (self) or transfer expressions from another subject (cross) by driving the Gaussians with FLAME parameters. Identity remains stable while expressions are well reproduced. The bottom row shows a free-form LBS-driven animation, where we drive the avatar through a continuous sequence of head poses and expressions. The avatar is temporally stable under moderate changes, while large movements can cause stretching or tearing. Because the input is a single image, parts not visible in it (e.g., teeth) are absent from the animation.

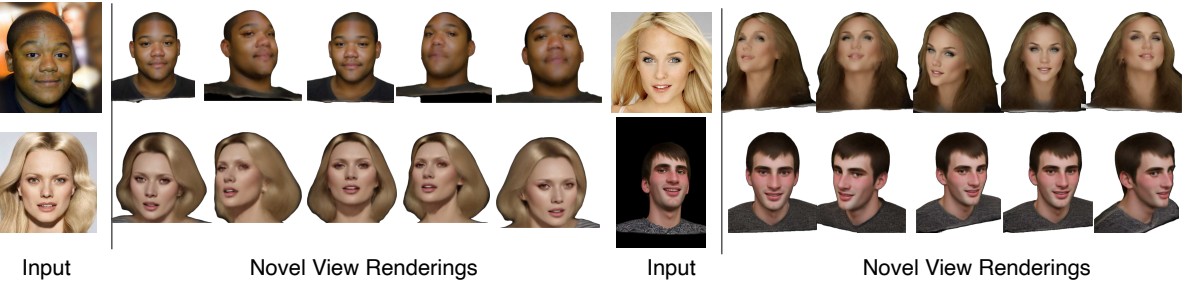

Figure 8: **In-the-wild reconstruction on FFHQ and CelebA-HQ.** For each of the four examples, the left image is a single real in-the-wild input, and the remaining columns are novel-view renderings of the reconstructed 3DGS model. FastAvatar recovers reasonable, identity-preserving avatars from many unconstrained images that differ substantially from the controlled NeRSemble capture setting.

## 4.5 Latent Space Structure and Decoder Analysis

We analyzed the latent space learned by the decoder to verify that it captures a smooth and generalizable representation rather than memorizing training identities. Fig. 9(left)) presents identity interpolation traversals between two latent codes. The intermediate reconstructions vary smoothly in geometry and appearance and remain consistent across novel viewpoints, indicating a smooth latent space. We also explored global attribute directions in the latent space (e.g., hair length, expression intensity). Traversing a code along these directions produces coherent edits while preserving identity and multi-view consistency (Fig. 9(middle, right)). We provide the procedure for estimating these directions and more results in Supplementary.

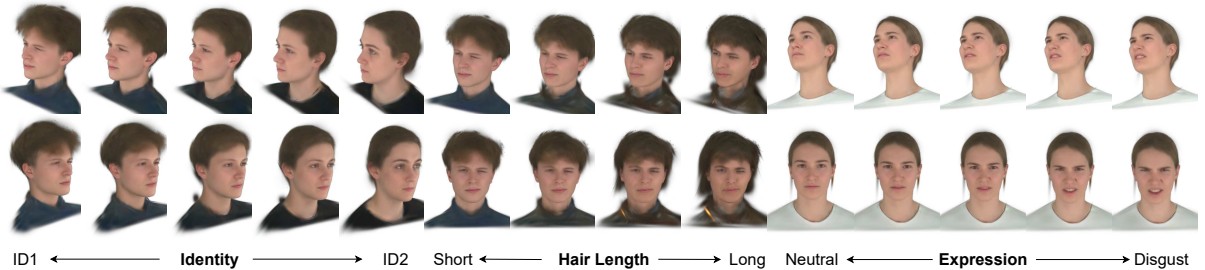

ID1 ← **Identity** → ID2    Short ← **Hair Length** → Long    Neutral ← **Expression** → Disgust

Figure 9: **Latent space interpolation and attribute traversals. Left:** Identity interpolation between two codes produces smooth, realistic transitions in geometry and appearance. **Middle and Right:** Moving along learned attribute directions (e.g., hair length, expression intensity) yields consistent edits across views while preserving identity, illustrating that the decoder has learned a meaningful latent space.

### 4.6 Template Analysis & Ablation Studies

#### 4.6.1 The Indispensability of the Averaged Canonical Template

A core contribution of our framework is the utilization of an averaged canonical template derived from the FLAME parametric model. Rather than predicting 3D Gaussian parameters (positions, scales, rotations, opacities, and spherical harmonics) entirely from scratch, our network predicts identity-specific residuals applied to this template. To validate the indispensability of this geometric prior, we analyze its impact on training convergence and structural stability against two specific baselines: (1) **Random Initialization**, where the base Gaussian attributes are initialized with random noise, and (2) **Joint Optimization**, where a global base template is learned from scratch jointly with the decoder network weights.

**Motivation: Constraining the Optimization Landscape.** The optimization landscape for 3D Gaussian Splatting is highly non-convex. When Gaussians are initialized randomly, the network must simultaneously solve for global topological structure, local scale, and color distribution. This often causes the optimization to fall into degenerate local minima, such as inverted opacities or severely oversized, misaligned splats. By initializing with a well-learned averaged canonical template, we constrain the geometric search space. The network is only burdened with learning subtle, identity-specific deformations, bypassing the chaotic early stages of geometric formation.

**Evidence 1: Training Convergence Efficiency.** The benefits of constraining the geometric search space are most evident during the training phase. As shown in Figure 10, our template-guided approach achieves rapid and highly stable training convergence. Because the averaged prior already provides a strong structural foundation, our method begins at a significantly lower reconstruction loss and smoothly descends to convergence within just 300 epochs. In contrast, baselines lacking this prior require substantially more training time (800 epochs) yet fail to reach comparable performance. Random initialization exhibits a slow, high-variance descent as the optimizer struggles to establish basic facial geometry from noise. Similarly, while joint optimization attempts to learn a global base structure, it plateaus at a noticeably higher loss. This demonstrates that simultaneously learning a base template and identity-specific deformations leads to an entangled optimization landscape, effectively capping the model's maximum reconstruction fidelity. Further quantitative analysis regarding the final reconstruction performance of these baselines is in Sec. 4.6.2.

**Evidence 2: Template Visualization.** To further validate the structural integrity of our prior, Figure 10(b) illustrates the construction and visual outcome of our canonical 3DGS template, $\mathcal{T}$. By systematically averaging the Gaussian parameters across models fit to our training distribution, the resulting template exhibits a smooth, continuous, and identity-neutral facial structure. It filters out high-frequency, subject-specific details while preserving essential, global anatomical topology. This visual evidence directly supports our intuition: rather than forcing the feed-forward network to synthesize geometry from a degenerate state, the template provides a physically plausible, mathematically stable anchor.

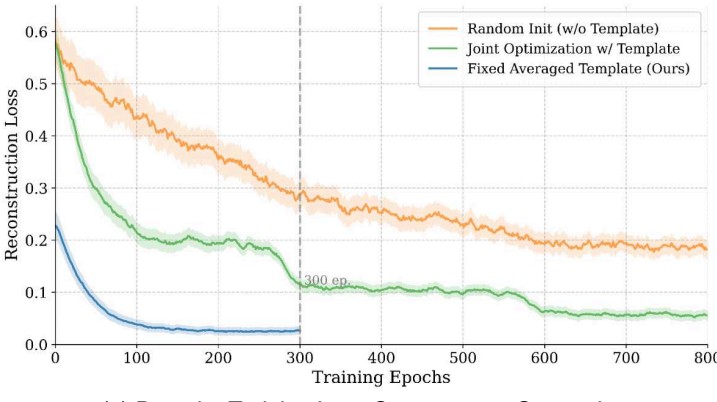
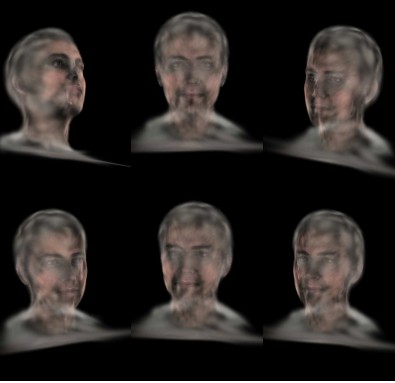

(a) Decoder Training Loss Convergence Comparison      (b) 3DGS Template Visualization

Figure 10: **(a) Decoder Training Loss Convergence Comparison**: Evaluation of the reconstruction loss during the optimization phase. Shaded regions denote the confidence bounds (variance) across 3 independent runs. By anchoring predictions to our *Fixed Averaged Template*, the optimization space is severely constrained, leading to highly stable, low-variance convergence within just 300 epochs. In contrast, simultaneously learning the base structure (*Joint Optimization w/ Template*) entangles the optimization landscape; the model gets trapped in local minima and exhibits higher variance before reaching a suboptimal plateau. Removing the geometric prior entirely (*Random Init w/o Template*) forces the optimizer to search for basic volume from noise, resulting in a highly unstable descent. **(b) 3DGS Template Visualization**: Visual representation of the averaged geometry that serves as the stable structural anchor, enabling the rapid and stable convergence observed in (a).

### 4.6.2 Ablation Studies

We finally present an ablation study of FastAvatar's key design choices in Table 2. We first find that fewer Gaussians ($K$=5023) lose detail, while a larger number ($K$=40,000) improves fidelity but slows inference. Our default $K$=10144 strikes a balance. Second, removing SSIM (decoder) or cosine distance (encoder) consistently degrades final rendering quality. Third, initialization with the learned template model yields the most stable and accurate reconstructions, outperforming joint-optimizing a canonical template during decoder training or random initialization. Fourth, test-time refinement improves results, though long runs (e.g., 600 steps) give diminishing returns relative to the temporal cost. Full results, including visual comparisons, are in Supplementary.

### 4.7 Stress Test of Inputs and Failure Cases

In real deployment, the input image rarely matches the clean laboratory capture of NeRSemble. To probe these conditions, we stress test FastAvatar on different types of difficult input: extreme input view, complex accessory such as a headscarf, non-ideal lighting, incomplete capture that crops part of the face, low-resolution image, and non-real face such as a statue or a stylized creature. Fig. 11 shows the input and the novel-view renderings for each case.

FastAvatar reasonably recovers geometry and identity for some of these inputs and fails for others. The hard cases reveal two characteristic failure modes. First, under an extreme yaw the input shows only one side of the face, so the model must hallucinate the occluded side. The unseen half is filled in from the hallucination prior learned by FastAvatar rather than from the

Table 3: Quality and identity consistency vs. input-view angular deviation from frontal. ID Sim. is the ArcFace cosine similarity between the rendered and ground-truth identity.

| Input dev. | PSNR↑ | SSIM↑ | ID Sim.↑ |
|---|---|---|---|
| 0–15° | 25.6 | 0.93 | 0.86 |
| 15–30° | 24.5 | 0.92 | 0.83 |
| 30–45° | 23.4 | 0.90 | 0.79 |
| 45–60° | 22.0 | 0.88 | 0.73 |
| > 60° | 20.1 | 0.85 | 0.66 |

true face, so it may be inaccurate and can produce a mild identity asymmetry between the two profiles. We quantify how this degradation grows with viewpoint in Table 3: because NeRSemble inputs differ from the frontal view along more than the yaw axis, we bin each input by its total angular deviation from the frontal camera, and report PSNR, SSIM, and identity similarity (ArcFace (Deng et al., 2019) cosine) per bin. All three drop smoothly as the deviation increases, indicating that identity consistency degrades gradually rather than collapsing under extreme yaw. Second, the test-time refinement optimizes toward a single image, so when the input is incomplete, low-resolution, or weakly tracked by the FLAME tracker, the refinement can amplify these defects rather than correct them. Non-real inputs such as the marble statue and the stylized creature fall outside the FLAME and face-recognition priors; FastAvatar still recovers a plausible head but loses fine appearance fidelity. These cases mark the current limits of single-image 3DGS reconstruction and motivate richer priors and tracker-aware refinement as future work.

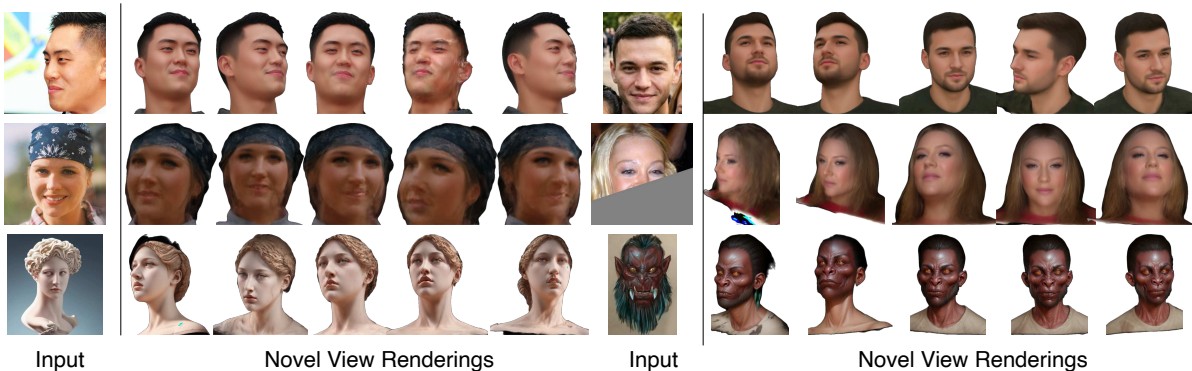

Input        Novel View Renderings        Input        Novel View Renderings

Figure 11: **Stress test on difficult inputs.** For each example, the left image is a single input and the remaining columns are novel-view renderings of the reconstructed 3DGS model. We test extreme input views, complex accessories, non-ideal lighting, incomplete captures, low-resolution images, and non-real faces (a statue and a stylized creature). FastAvatar reasonably recovers some inputs and fails on others: extreme yaw can cause mild identity asymmetry on the hallucinated occluded side, and non-real or weakly tracked inputs reduce appearance fidelity. For some inputs where the FLAME tracker fails to produce a reliable fit, we omit the test-time refinement or reduce its number of steps.

We also look at varied lighting and non-neutral expressions. Since natural images carry no explicit lighting control, we relight the inputs with a single-image portrait relighting method (Zhou et al., 2019) and test whether FastAvatar separates lighting from identity or bakes it in. As Fig. 12 shows, FastAvatar bakes the lighting in: the novel views still carry the input lighting, though not always precisely. Without intrinsic decomposition and with a fixed Gaussian budget for real-time speed, the network captures illumination only within the appearance parameters and cannot resolve high-frequency specularities. It also handles different expressions, though tooth modeling remains a separate problem.

## 5 Discussion and Conclusion

**Limitations.** FastAvatar inherits structural constraints from FLAME and the face-recognition backbone used by the encoder. FLAME does not model long hair, fine strands, or clothing, and most face-recognition models crop tightly around the face. As a result, subjects with long hairstyles, prominent bangs, hats, or high-variation clothing, especially common in female subjects, may exhibit smoothed or incomplete reconstructions around these regions; we show some failure cases in Fig. 11 and more in the Supplementary. These limitations are shared by most current 3DGS-based avatar systems and point toward the need for more expressive priors and broader training data. In addition, the Nersemble dataset itself has limited demographic and appearance diversity, which constrains the range of identities and hairstyles that current methods can reliably model. Richer multi-view datasets would help explore the full capabilities and limitations of single-image 3DGS reconstruction in this space.

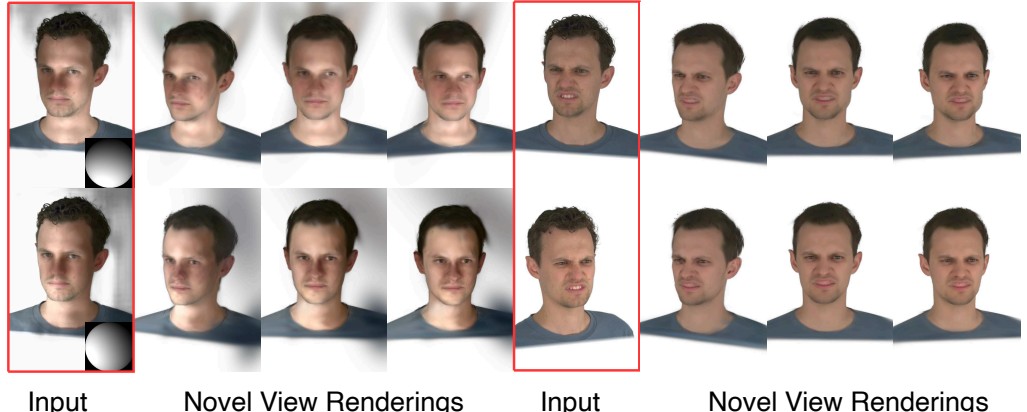

Input   Novel View Renderings   Input   Novel View Renderings

Figure 12: **Lighting and expression stress.** We generate inputs under different lighting with single-image portrait relighting (Zhou et al., 2019). The input is in the red box (with an inset light-direction probe), and the remaining columns are novel-view renderings under varied illumination and non-neutral expressions. FastAvatar bakes the lighting into the representation, so the novel views still carry the input lighting effects, though not always precisely, while high-frequency specularities and tooth regions remain harder to resolve.

FastAvatar provides a practical solution for single-image 3D face reconstruction across large pose variations. The two-stage pipeline: an encoder–decoder feed-forward prediction followed by a lightweight refinement, allows us to recover clean geometry in one pass and improve appearance with only $\sim 3$ seconds of additional computation. First, a key component of this design is the mesh-attached average Gaussian template, which establishes stable semantic correspondences across subjects and makes residual prediction well conditioned. Our ablation studies confirm that this canonical parameterization is critical for reliable geometry estimation and avoids the instability. The encoder–decoder architecture further contributes to generalization. By encouraging a pose-invariant latent representation during training, the model can handle a wide variety of inputs and maintain identity consistency for unseen subjects. Together, these components enable FastAvatar to produce high-quality reconstructions while keeping the overall system simple and efficient.

Overall, FastAvatar provides a simple, robust, and efficient framework for real-time single-image 3D face reconstruction, substantially improving the quality–speed trade-off in 3DGS avatars and offering a practical foundation for interactive and downstream animation applications.

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

# A  Implementation Details

## A.1  Dataset

We conducted experiments on Nersemble, a large-scale face dataset containing multi-view images across diverse identities and expressions. Specifically, our dataset includes *422 subjects*, each captured under *16 different camera viewpoints*, with controlled variations in pose, expression. Each subject has about *100 images*, yielding around *40k training samples* in total.

To enable high-quality 3D Gaussian Splatting reconstructions, we extracted camera intrinsics and extrinsics using VHAP (Qian, 2024). Meshes were initialized from the FLAME parametric face model, and subsequently converted into Gaussian primitives.

**Sample Images:** See Figure 13 for sample images of Nersemble.

## A.2  Preprocessing

We performed the following preprocessing steps:

- **Resizing**: Input images were resized to $802 \times 550$.

- **Gaussian initialization**: Each subject's mesh was rasterized into 10144 Gaussians, with position, scale, rotation, and opacity initialized from mesh geometry.

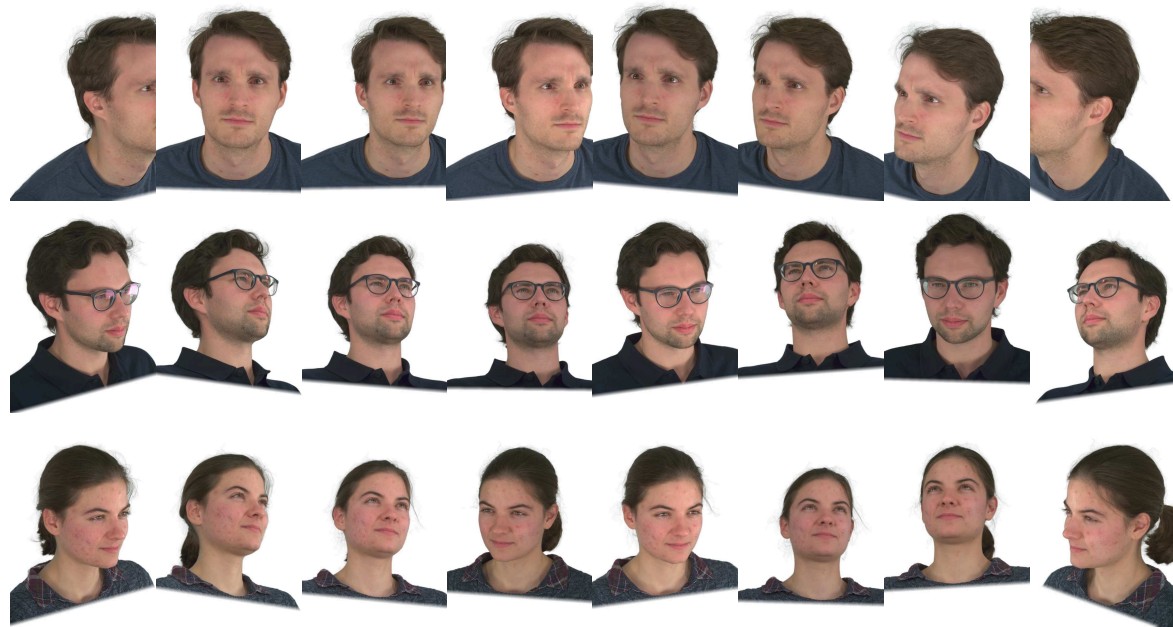

Figure 13: **Sample images from Nersemble.** Each row shows different subjects under varying head poses and facial expressions, demonstrating the diversity in identity, viewpoint, and expression that our method is designed to handle.

Table 4: Summary of FastAvatar model architecture.

| Component | Input Dim | Hidden Layers | Output Dim | Purpose |
|---|---|---|---|---|
| Shared Body (Encoder) | $512 + 32$ | $6 \times 256$ | 256 | Joint latent–Gaussian features |
| Scale Head | 256 | $1 \times 256$ | 3 | Anisotropic scale updates |
| Rotation Head | 256 | $1 \times 256$ | 4 | Quaternion parameters |
| Color Head | 256 | $1 \times 256$ | 3 | SH color coefficients |
| Opacity Head | 256 | $1 \times 256$ | 1 | Opacity residuals |
| Gaussian Embeddings | – | – | 32 | Per-Gaussian learnable embeddings |
| View-Invariant Encoder | $3 \times H \times W$ | 4 MLP layers | 512 | Pose-invariant latent code |

## A.3 Model Architecture

FastAvatar's backbone model is composed of three key components: (i) a *mlp encoder* for Gaussian parameter modulation, (ii) a *base Gaussian field* initialized from precomputed splats, and (iii) a *view-invariant encoder* for pose-robust latent code extraction. Together, these modules enable dynamic conditioning of Gaussian primitives on both latent identity codes and local geometric embeddings.

### A.3.1 MLP Encoder for Conditional Modulation

The encoder is a multi-layer perceptron (MLP) designed to predict residual updates to Gaussian parameters, conditioned jointly on a latent code $w \in \mathbb{R}^{512}$ and a Gaussian embedding $g \in \mathbb{R}^{32}$. The concatenated input vector is processed by a *shared body network* followed by *specialized heads* for each parameter type. The shared body consists of 6 fully connected layers with ReLU activations, each with hidden dimensionality of 256. Five parameter-specific heads branch from this shared representation:

- **Scale head** ($\mathbb{R}^3$): predicts anisotropic 3D scaling.

- **Rotation head** ($\mathbb{R}^4$): outputs quaternion updates.

- **Color head** ($\mathbb{R}^{d_{SH}}$): produces spherical harmonics coefficients, split into a DC component ($sh_0$) and higher-order terms ($sh_N$).

- **Opacity head** ($\mathbb{R}^1$): predicts per-Gaussian opacity.

- **Means head** ($\mathbb{R}^3$): predicts residual 3D displacements.

All heads use a 2-layer MLP (Linear–ReLU–Linear) with Xavier initialization for stability. The MLP therefore enables fine-grained, differentiable modulation of Gaussian primitives.

### A.3.2 Base Gaussian Field

We initialize the scene with a set of *base Gaussians* derived from a precomputed PLY file. Each Gaussian stores a quaternion rotation, anisotropic scale, spherical harmonics coefficients, and opacity. These base parameters remain fixed during training. Each Gaussian is also assigned a *learnable embedding vector* ($\mathbb{R}^{32}$), initialized from normalized positions and refined during training.

### A.3.3 View-Invariant Encoder

To enforce pose invariance, we employ a face encoder inspired by Arcface (Deng et al., 2019) models. Input face images are mapped into a 512-dimensional latent space, invariant to pose and lighting variations. A projection head then refines these features via a 4-layer MLP:

$$512 \;\rightarrow\; 1024 \;\rightarrow\; 768 \;\rightarrow\; 512 \;\rightarrow\; 512,$$

with batch normalization, ReLU activations, and dropout regularization. A learnable temperature parameter is applied to rescale embeddings, enabling adaptive similarity calibration.

### A.3.4 Architecture Summary

Table 4 summarizes the model components.

### A.4 Training Details

- **Loss functions**: As described in expression (7) - (9) in main text, we choose the following hyper-parameters: $\lambda_1 = 0.1, \lambda_2 = 0.6, \lambda_3 = 0.3, \lambda_4 = \lambda_5 = 5e - 3, \lambda_{cos} = 0.1$.

- **Optimizer**: We use Adam optimizer for all the training with the following learning rates: $lr_{decoder} = 1e - 4, lr_{embedding} = 5e - 3, lr_{idcode} = 5e - 3, lr_{encoder} = 1e - 3$. We also use ReduceLROnPlateau learning rate schedulers for all optimizers, $factor = 0.5, patience = 5$.

- **Batch size**: 1 for decoder training, 64 for encoder training.

- **Training epochs**: 300 epochs for both encoder and decoder training.

- **Hardware**: All experiments were conducted on one NVIDIA A100 GPU. Decoder training takes around $8 hours$, encoder training takes around $6 hours$.

## B Camera Pose Estimation

We estimate test-time camera poses for frozen 3D Gaussian Splatting (3DGS) human avatars to support viewpoint-matched rendering and fair quantitative evaluation. Single-image avatar methods such as Arc2Avatar (Gerogiannis et al., 2025) produce consistent geometry, but do not provide poses for unseen inputs, making direct PSNR/SSIM comparison difficult. We therefore recover poses for test images with a simple three-stage procedure.

### B.1 Camera Initialization

For every training and test view, we initialize the camera using the VHAP-based fitting of Qian (2024), adopted in GaussianAvatars (Qian et al., 2024a). In brief, FLAME (Li et al., 2017b) is aligned to each image from 2D facial landmarks while fixing identity and optimizing translation, head rotation, and expression. This yields a coarse yet stable starting pose $(R_i, \mathbf{t}_i)$ per image.

**Frozen model setting.** Let $G = \{\mu_k, s_k, q_k, \alpha_k, c_k\}_{k=1}^K$ denote the 3DGS parameters. Across all stages, $G$ remains fixed (no densification/pruning/splitting/opacity resets); only camera parameters are updated.

**Canonical alignment.** We begin with the per-view extrinsics $(R_i, \mathbf{t}_i)$ from VHAP and align the entire set to a shared frontal reference. NeRSemble provides a calibrated forward-facing multi-camera rig (16 views). We use the most front-facing rig camera which is commonly treated as the frontal view in prior NeRSemble-based evaluations as our frontal anchor (Kirschstein et al., 2023); taking its VHAP extrinsics gives $R_{\text{front}}$. Many single-image baselines define their own canonical "front" camera for rendering; we extract this reference rotation $R_{\text{target}}$ from the baseline's official code (Gerogiannis et al., 2025). The global alignment is

$$R_g = R_{\text{target}} R_{\text{front}}^\top, \tag{3}$$

which we project to SO(3) and apply to every view:

$$R_i \leftarrow R_g R_i. \tag{4}$$

### B.2 Global Shift Optimization

**Translation-only hill climbing.** Keeping $R_i$ fixed, we refine translations by random hill climbing. At iteration $t$, we sample a view $i$ and propose

$$\mathbf{t}_i^{\text{cand}} = \mathbf{t}_i + \boldsymbol{\epsilon}_t, \qquad \boldsymbol{\epsilon}_t \sim \mathcal{N}(\mathbf{0}, \sigma_t^2 I_3), \tag{5}$$

with exponentially decaying $\sigma_t$. We accept the update if it lowers a simple foreground IoU loss between the rendered avatar and the image; otherwise we keep $\mathbf{t}_i$.

### B.3 Pose Refinement

**Residual 6-DoF refinement.** Finally, we optimize per-view 6-DoF camera residuals through the differentiable renderer, following GaussianHaircut (Zakharov et al., 2024) with BARF-style parametrization (Lin et al., 2021), but with $G$ frozen. We use a face-restricted photometric loss: a frozen LangSAM segmenter (Medeiros, 2023) with prompt `face` provides mask $M_i = S(I_i)$, and we minimize

$$\mathcal{L}_{\text{Stage C}} = \left\| \hat{I}_i \odot M_i \; - \; I_i \odot M_i \right\|_1, \tag{6}$$

where $\hat{I}_i = \mathcal{R}(G, R_i, \mathbf{t}_i)$. This refines poses to best match facial appearance without changing the avatar.

## C Multi-pose & Multi-view Results

A key strength of our approach lies in its ability to produce reconstructions that remain consistent across diverse pose inputs. We compare feed-forward predictions (base) with the results obtained after full optimization refinement in Figure 15, 16, 17. While the feed-forward pipeline already delivers stable geometry and appearance under varying viewpoints, the optimization stage further improves fine-scale alignment and visual fidelity. Across multiple subjects, our method maintains coherent identity and expression even under challenging pose variations, demonstrating robustness to camera viewpoint changes.

| Method | DiffusionRig | GAGAvatar | Arc2Avatar | FlashAvatar | GaussianAvatars | LAM | Ours(base) | Ours(full) |
|--------|--------------|-----------|------------|-------------|-----------------|-----|------------|------------|
| ID Sim. ↑ | 0.65 | 0.76 | 0.61 | 0.35 | 0.39 | 0.77 | 0.70 | **0.83** |

Table 5: Identity similarity scores across methods, measured using cosine similarity of Arcface (Deng et al., 2019) embeddings (higher is better), and a typical threshold used to identify where a query face and a target face is $t = 0.6$. Our method preserves subject identity, and our full optimization results achieve the best scores among all the baselines.

## D   Identity and Attribute Manipulation

A key advantage of FastAvatar's learned representation is its well-structured latent space which supports real-time identity and attribute manipulation. Importantly, the success of these manipulations provides evidence that our decoder has learned a generalizable, continuous mapping from latent codes to 3DGS parameters, rather than merely memorizing the training identities. The smooth transitions we observe when interpolating between identities or editing attributes demonstrate that the decoder can synthesize novel faces not present in the training set, validating our multi-stage training strategy.

For example, to interpolate between two identities encoded by embeddings $w_1$ and $w_2$, we can simply perform linear interpolation: $\tilde{w} = (1 - \alpha)w_1 + \alpha w_2$ with $\alpha \in [0, 1]$. The resulting intermediate identities are photorealistic and anatomically plausible, indicating that the latent space is continuous and well-structured.

To edit a facial attribute (e.g., expression, hair length), we can use a latent space traversal technique most famously used with GANs (Radford et al., 2015; Shen et al., 2020; Liang et al., 2023). For a given binary face attribute $y \in \{-1, +1\}$, we obtain a pretrained classifier to obtain a binary label per subject (the continuous case is also easily handled with a regression model). We then train a linear Support Vector Machine (SVM) to find the optimal hyperplane separating positive and negative attribute samples in the latent space. The normal vector of the hyperplane $\hat{\mathbf{n}} = \mathbf{n}/\|\mathbf{n}\|$ encodes a direction in latent space highly correlated with the target attribute. To edit the attribute, we can simply traverse along this direction: $\tilde{w} = w + \lambda\hat{\mathbf{n}}$, where $\lambda$ controls the magnitude of the change. The direction $\hat{\mathbf{n}}$ is *global* (i.e., not identity-specific), enabling meaningful edits across subjects and poses, further demonstrating the generalizability of our learned representation.

As shown in Figure 19, interpolations between subject embeddings produce smooth transitions across identities while preserving facial structure and pose consistency. Similarly, expressions edits 20, 21, can be achieved via latent space traversal, resulting in coherent modifications across multiple viewpoints. Notably, the generated results remain consistent with respect to varying rendering views, demonstrating that both identity interpolation and attribute manipulation generalize reliably beyond the training distribution.

## E   Identity Similarity Score

As mentioned in Figure 3 in main text, we further assess how well each method preserves subject identity by measuring cosine similarity of Arcface (Deng et al., 2019) embeddings between the reconstructed and ground-truth images. Table 5 shows that our approach achieves substantially higher identity similarity compared to prior methods. In particular, our full optimization reaches 0.83, well above the commonly used verification threshold of t = 0.6, and clearly outperforming baselines such as Arc2Avatar (0.61) and FlashAvatar (0.35). These results highlight FastAvatar's ability to maintain subject identity even under unconstrained poses, with consistent improvements in both feed-forward and full refinement modes.

## F   Full Ablation Study

We conduct a comprehensive ablation study to analyze the impact of key design choices in FastAvatar. Table 6 reports PSNR, SSIM, LPIPS and runtime across variations in Gaussian count, loss functions, initialization strategies, and refinement iterations. The results highlight that under-parameterization (5k Gaussians) severely degrades reconstruction quality, while increasing Gaussians beyond 20k yields diminishing

| Setting | PSNR ↑ | SSIM ↑ | LPIPS ↓ | Runtime (ms) |
|---|---|---|---|---|
| *Number of Gaussians K (Default: 10,144)* | | | | |
| 5023 | 20.94 | 0.84 | 0.42 | 2800 |
| 20,000 | 24.85 | 0.92 | 0.19 | 3000 |
| 40,000 | 25.58 | 0.93 | 0.17 | 8500 |
| *Loss Weights (Default: Eq. (7)–(9))* | | | | |
| w/o SSIM (dec.) | 22.03 | 0.88 | 0.25 | 3000 |
| w/o CosSim (enc.) | 22.27 | 0.89 | 0.24 | 3000 |
| *Gaussian Init. (Default: Average Template)* | | | | |
| Random Init | 21.92 | 0.88 | 0.33 | 3000 |
| Joint-Opt. | 22.84 | 0.89 | 0.28 | 3000 |
| *Refinement Iterations (Default: 300)* | | | | |
| 0 (feed-forward) | 21.17 | 0.89 | 0.20 | 10 |
| 100 | 22.93 | 0.91 | 0.19 | 1000 |
| 600 | 24.21 | 0.91 | 0.18 | 6000 |
| **FastAvatar (full)** | **24.09** | **0.92** | **0.18** | **3000** |

Table 6: Extended ablation study on FastAvatar design choices. We report PSNR, SSIM, LPIPS, and runtime. Our full model combines average-template initialization, full loss, $K = 10,144$ Gaussians, and 300 refinement iterations.

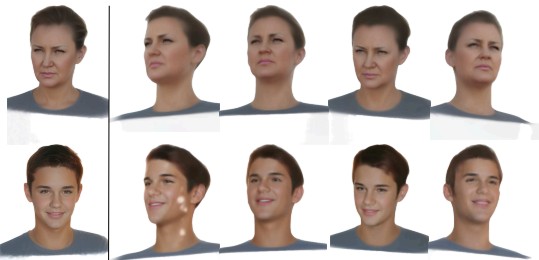

Figure 14: **Generalization to additional out-of-distribution identities.** Left column = single input image. The remaining columns = novel-view renderings from the reconstructed 3DGS model.

returns. Both SSIM and cosine-similarity losses contribute noticeably to perceptual fidelity, and canonical template initialization proves substantially more effective than random or joint optimization. Finally, test-time refinement progressively improves quality, though perceptual metrics benefit more, which saturates early. Together, these results validate our architectural and training design choices as critical to achieving high-fidelity, real-time reconstructions.

## G  Additional OOD Identities Results

To evaluate generalization to identities not present in the training set, we generate a diverse set of out-of-distribution (OOD) test subjects using a privacy-preserving pipeline based on ControlNet-driven (Zhang et al., 2023) diffusion synthesis. This allows us to create photorealistic yet entirely synthetic identities that do not correspond to any real person, ensuring privacy while enabling rigorous evaluation across a wide spectrum of facial structures, skin tones, hairstyles, and age groups.

## H  Failure Case Study

Although our method performs robustly across most subjects, several consistent failure modes emerge in challenging scenarios. Below we highlight the most common categories and analyze their underlying causes.

**Long Hair, Facial Hair and Volumetric Hairstyles.** Subjects with long, curly, or high-volume hair/beard present the most severe failure cases, and example can be found in Figure 17, 18. Because

our Gaussian template is aligned to a FLAME head topology, it contains no explicit representation of volumetric hair. As a consequence, long hair often collapses toward the scalp under large viewpoint changes, while curls or strands become over-smoothed or drift across views. These errors stem from a *structural limitation*: high-volume hair requires an explicit hair model or an additional volumetric layer, which cannot be captured by face-aligned Gaussian indices alone.

**Aged Subjects.** Elderly individuals are underrepresented in the training distribution, leading to noticeable artifacts such as missing wrinkles, weakened nasolabial folds, and inaccurate skin reflectance. Since the FLAME expression space and the Gaussian SH bases predominantly encode smooth variations, age-specific high-frequency details are regressed toward the dataset mean, causing older faces to appear unusually youthful or oversmoothed.

**Lighting Variations.** Our model assumes that moderate illumination changes can be absorbed by Gaussian SH coefficients. However, out-of-distribution lighting, including strong side lighting, high-contrast shadows, or colored illumination, can cause inconsistent SH estimates and instability in opacity or scale. These patterns reflect the limited lighting diversity of the training set, which is dominated by uniformly lit captures.

**Clothing and Shoulder Regions.** When upper-body regions enter the capture, clothing with strong textures or patterns is often oversmoothed and inconsistent across views. These artifacts arise from the limited variation in clothing within the training set and from the fact that the canonical Gaussian scaffold is optimized primarily for facial geometry.

**Conclusion.** Across these categories, most failure modes arise from a combination of *limited or imbalanced training data* and the *inherent bias of the FLAME-based canonical scaffold*. Groups such as elderly subjects, individuals with long hair or dense beards, and scenes with non-neutral illumination appear infrequently in the training set, resulting in underconstrained priors and reduced generalization. In contrast, long hair and other volumetric structures represent a more fundamental limitation: they cannot be faithfully reconstructed without an *explicit hair model* or an additional volumetric layer. Addressing these structural and dataset-related issues offers a clear path toward improving robustness in future work.

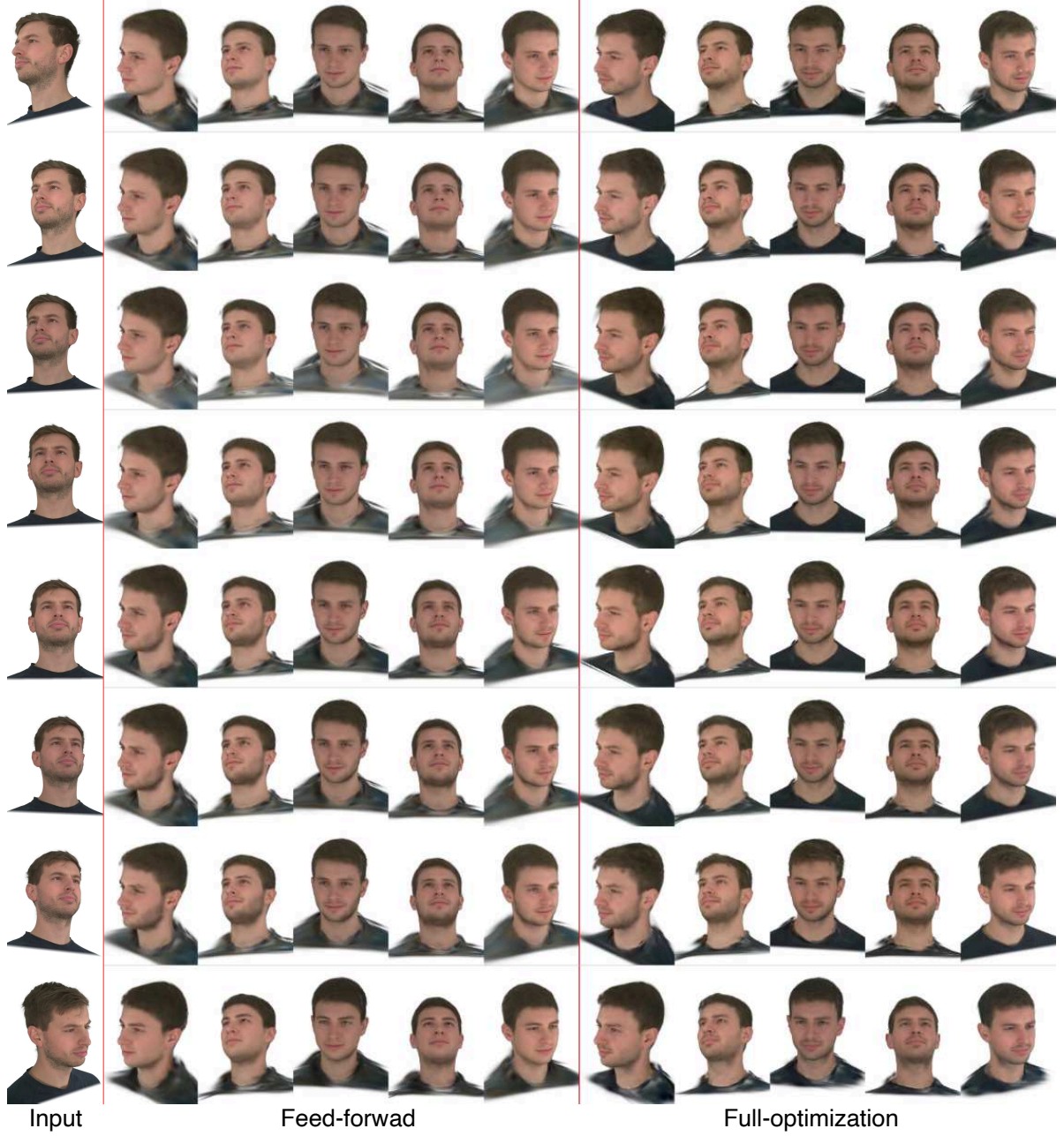

Input            Feed-forwad            Full-optimization

Figure 15: **Multiple input poses, multiple rendering views results for subject 1.** Each row corresponds to a single subject using different view poses. From left to right: input image, feed-forward reconstruction, and full optimization result. The refinement step improves fine-scale alignment and visual fidelity while maintaining consistency across views.

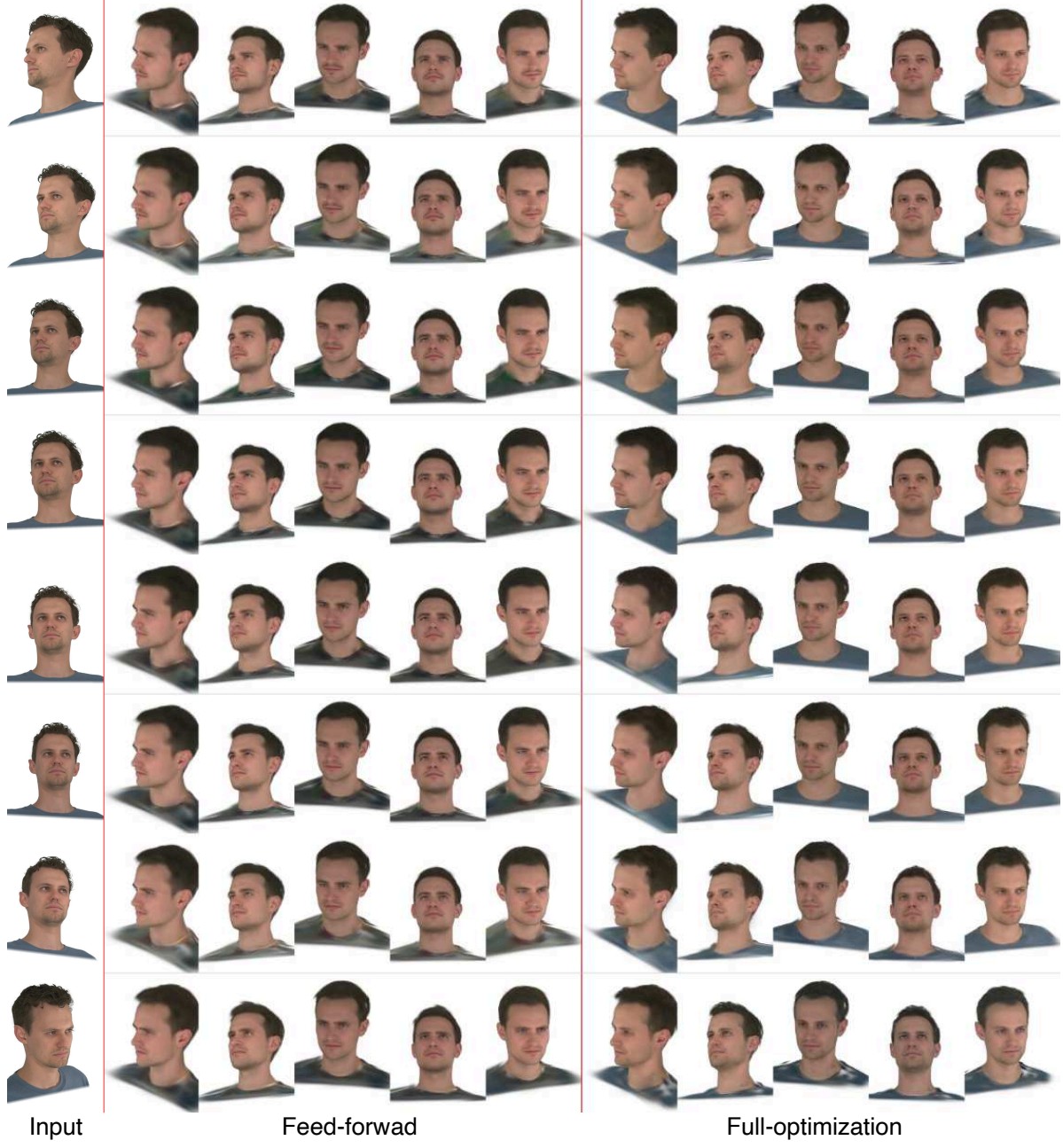

Input               Feed-forwad               Full-optimization

Figure 16: **Multiple input poses, multiple rendering views results for subject 2.** Each row corresponds to a single subject using different view poses. From left to right: input image, feed-forward reconstruction, and full optimization result. The refinement step improves fine-scale alignment and visual fidelity while maintaining consistency across views.

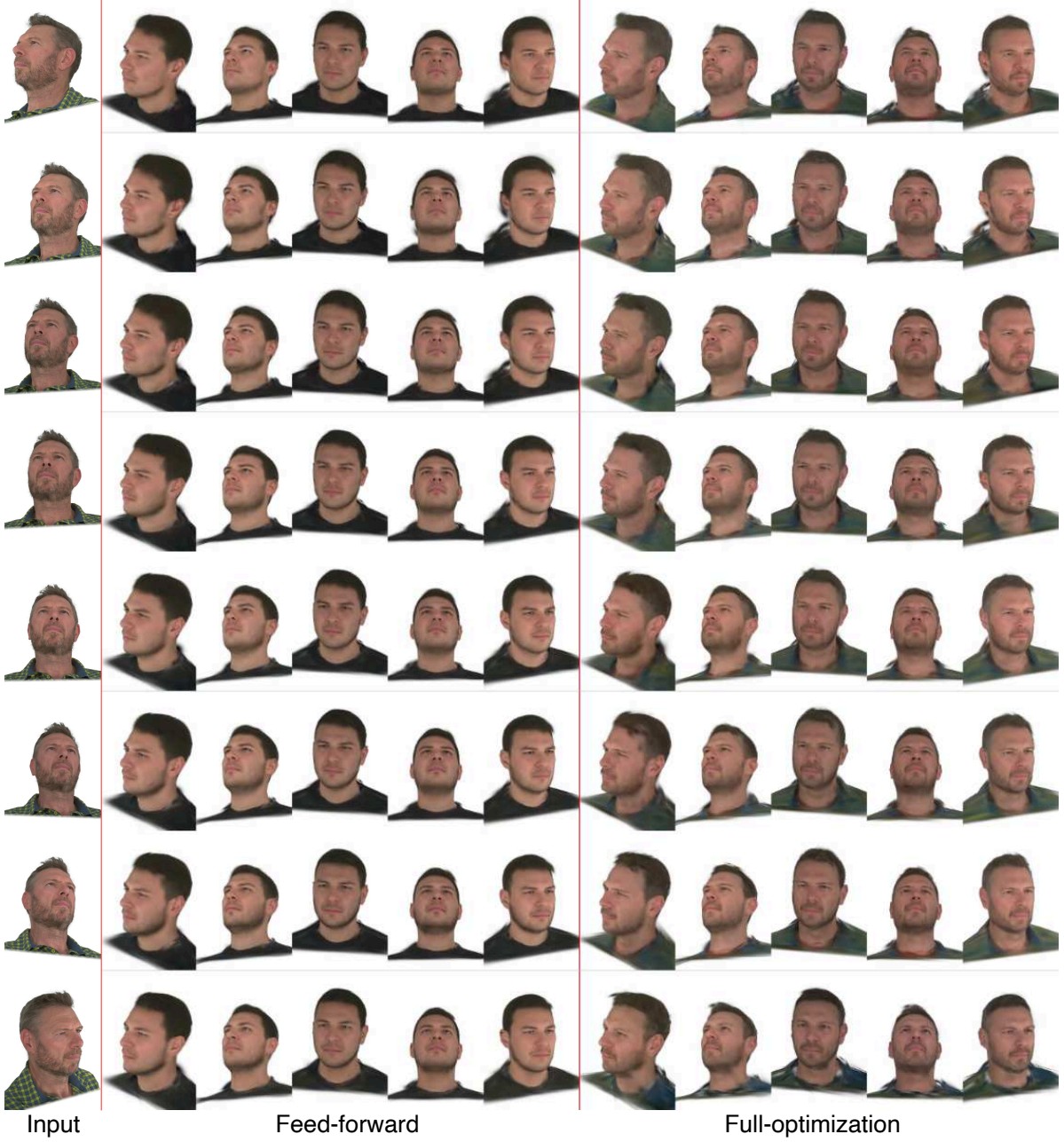

Input          Feed-forward          Full-optimization

Figure 17: **Multiple input poses, multiple rendering views results for subject 3.** Each row corresponds to a single subject using different view poses. From left to right: input image, feed-forward reconstruction, and full optimization result. The refinement step improves fine-scale alignment and visual fidelity while maintaining consistency across views.

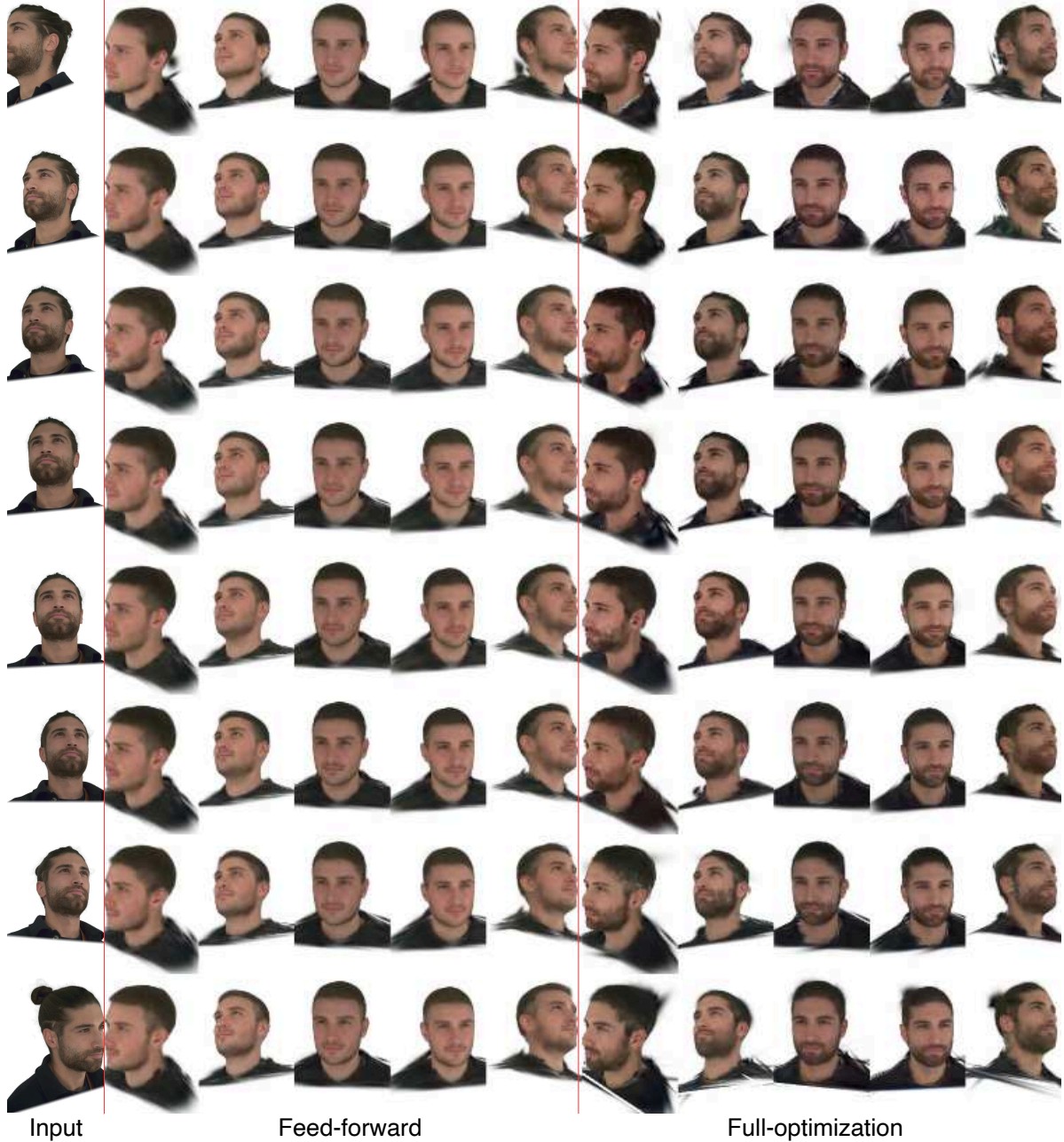

Input     Feed-forward     Full-optimization

Figure 18: **Multiple input poses, multiple rendering views results for subject 4 (failure case).** Each row corresponds to a single subject using different view poses. From left to right: input image, feed-forward reconstruction, and full optimization result. The refinement step improves fine-scale alignment and visual fidelity while maintaining consistency across views. Our model fails to reconstruct the long hair.

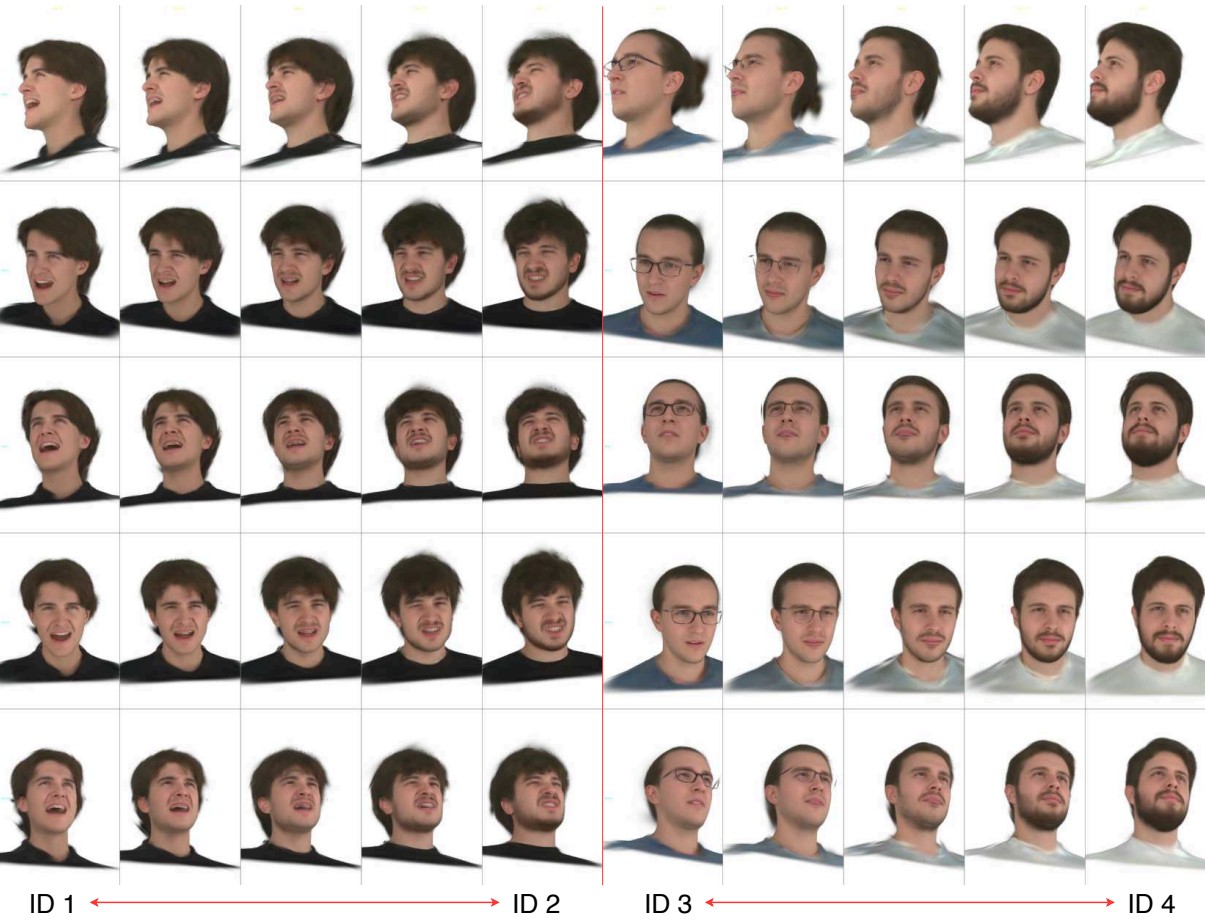

ID 1 ←――――――――――――――――→ ID 2    ID 3 ←――――――――――――――――→ ID 4

Figure 19: **Identity traversal.** Similar to results shown in Fig. 4.3 in main text, we show results on identity traversal on more subjects and rendering views. These results show a consistent and smooth transform between two identities. Note the minor expression changes, which FastAvatar handles robustly..

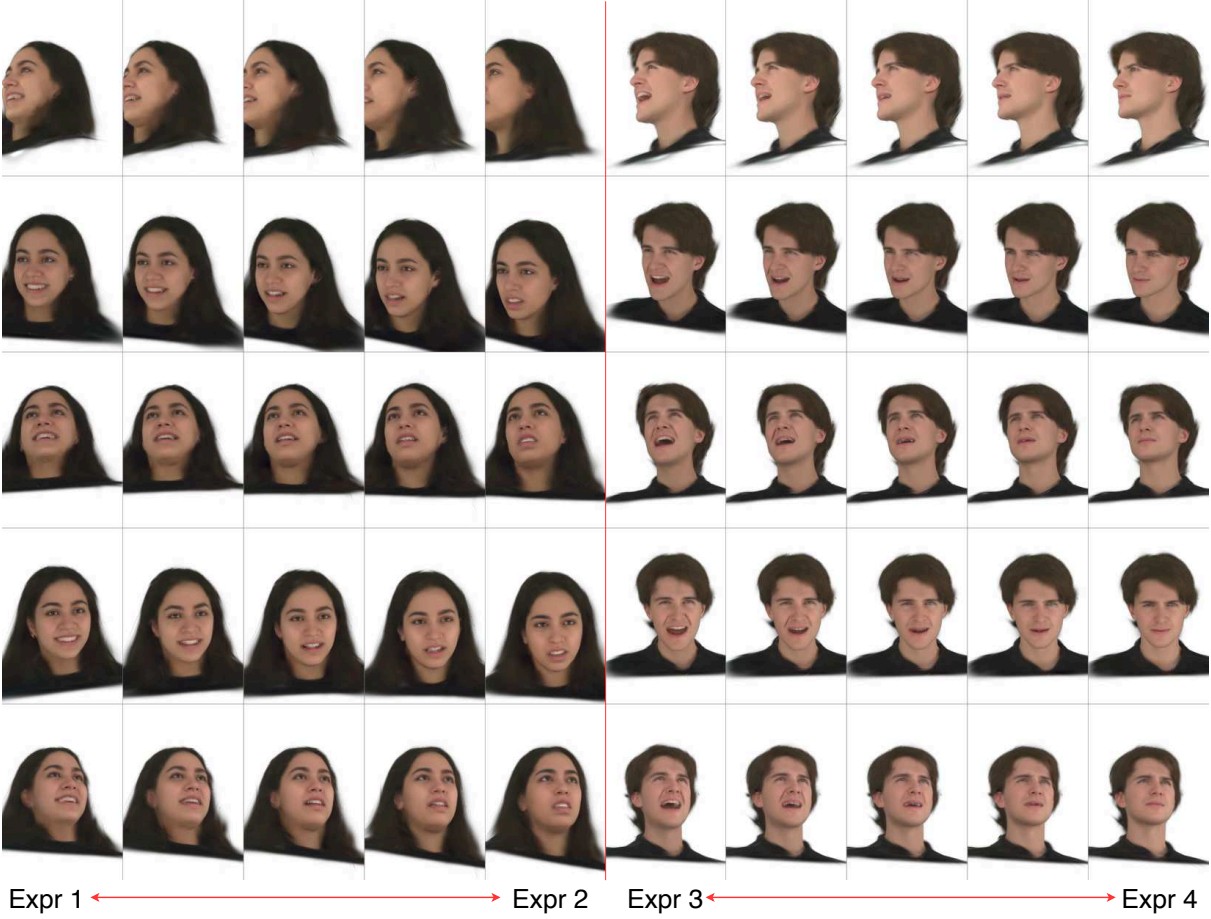

Figure 20: **Real-time expression manipulation part 1.** We show results on real-time expression manipulation using method described in Section 4.3 in main text. FastAvatar is capable of manipulating expression in real-time without modifying other facial features.

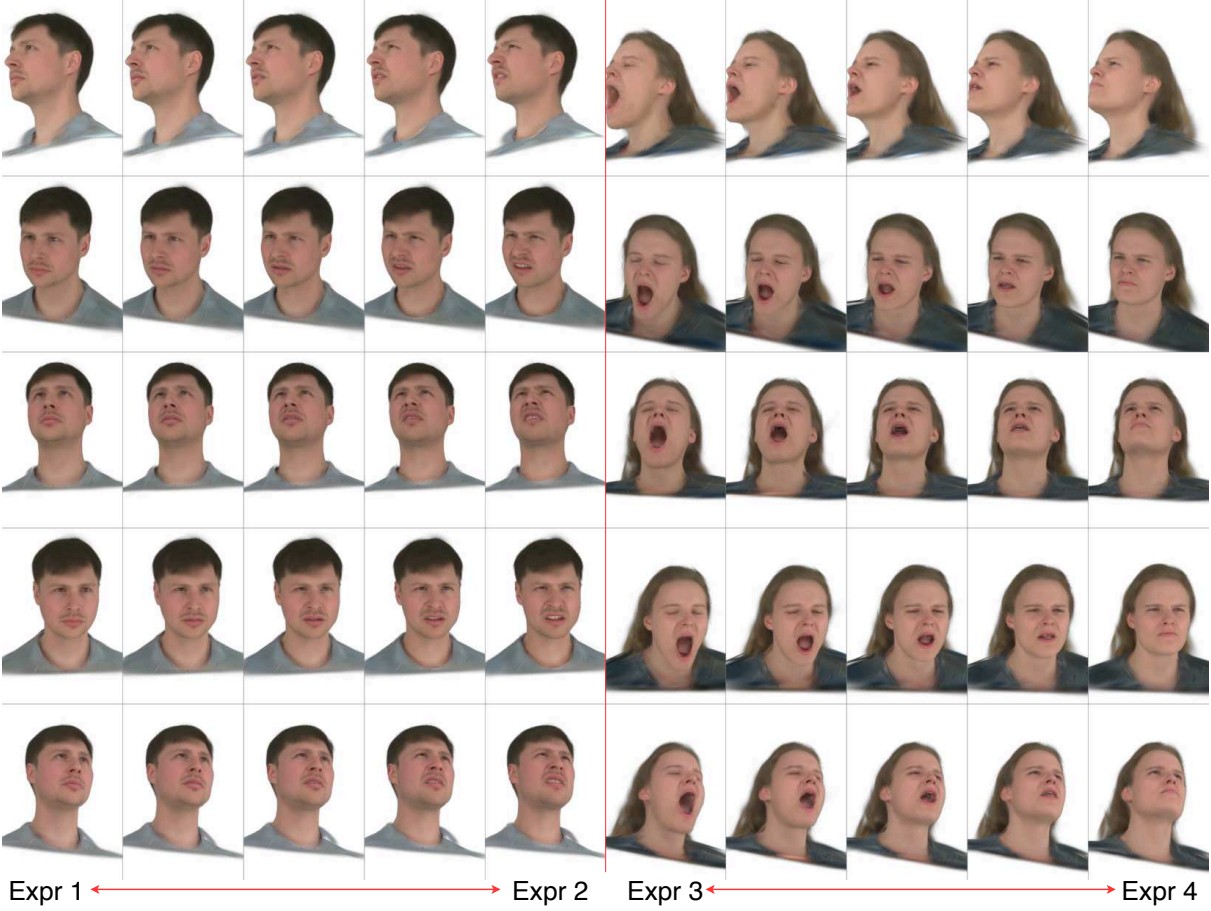

Expr 1 ←——————————————————→ Expr 2    Expr 3 ←——————————————————→ Expr 4

Figure 21: **Real-time expression manipulation part 2.** We show results on real-time expression manipulation using method described in Section 4.3 in main text. FastAvatar is capable of manipulating expression in real-time without modifying other facial features.

