# OpenReview forum: "FastAvatar: Rapid 3D Gaussian Splatting Face Avatar Generation from a Single Image"
_TMLR — Under review for TMLR_

### Review · Reviewer_3LEy · 2026-05-03

**Summary Of Contributions:**

This paper presents "FastAvatar," a highly efficient framework for single-image 3D face avatar generation. By anchoring 3D Gaussian Splatting (3DGS) parameters to an averaged, FLAME-based canonical template and predicting residuals, the authors successfully tackle the severe speed bottlenecks of optimization-based 3DGS methods while avoiding the structural instability common in feed-forward networks. The two-stage pipeline (feed-forward prediction followed by a lightweight latent refinement) is clever, well-motivated, and yields impressive empirical results (e.g., 24.01 dB PSNR in ~3 seconds) compared to state-of-the-art baselines.

**Audience:**

Yes

**Audience Explanation:**

Although I am not an expert in this field, the technical solution is reasonable.

**Claims And Evidence:**

No

**Claims Explanation:**

The manuscript suffers from several clarity issues regarding mathematical notation, and some of the inherent limitations of the methodology could be framed more transparently in the main text. Please see the "Requested Changes" section for details.

**Requested Changes:**

Please note that my expertise is outside the specific subfield of 3D Gaussian Splatting (3DGS) and neural rendering. Consequently, I cannot definitively judge the rigorousness of the empirical evaluation, the choice of baselines, or the hyperparameter tuning. My comments and suggestions below are provided from the perspective of a general machine learning practitioner, with a focus on improving the paper's clarity, structural presentation, and accessibility for a broader audience.

The manuscript introduces several mathematical formulations and terms without clear, immediate definitions, making it difficult for readers unfamiliar with the deepest nuances of 3DGS to follow:
* The $(K \times P)$ Parameter Block: At the end of Section 2.2, the text mentions "removing the $(K \times P)$ parameter block." However, $K$ and $P$ are never defined in this section. While $K$ is later defined in Section 3.1 as the number of Gaussians, $P$ is never explicitly defined as the "number of parameters per Gaussian" anywhere in the main text. From the perspective of a broader audience, it is difficult to understand the scale and potential impact of these parameters.
* Undefined Symbols in Section 3: In Section 3.1, the rasterization equation introduces variables $D$, $d$, and $j$ ($C = \sum_{d=1}^D c_d \alpha_d \prod_{j<d}(1-\alpha_j)$). While recognizable to 3DGS domain experts as ray-marching depth indices, these must be explicitly defined for a general machine learning audience. Similarly, the dimensions and precise nature of the expressions in Section 3.3.1 (e.g., the $\lambda$ weighting terms) and Section 3.5 could be defined much more rigorously. Similarly, the details definition of the encoder and decoder should be provided.

In Section 3.3.1, the authors mention initializing Gaussian embeddings using "sinusoidal positional encodings of their canonical FLAME coordinates." However, what these canonical coordinates actually represent (e.g., the 3D Cartesian coordinates of the unposed mesh's face centers) is not explicitly defined in the text, relying entirely on assumed domain knowledge and citations.

In the introduction, the authors critique existing methods for requiring "multi-view captures," positioning FastAvatar as a solution because it uses a single image. However, the proposed template and training pipeline heavily rely on the NeRSemble multi-view dataset. While it is understood that the inference stage is strictly single-view, the distinction between the rigorous multi-view requirements for training the prior versus the single-view requirement for inference is somewhat blurred early in the text. Clarifying this distinction upfront would prevent reader confusion.

---

> ### Author Response · Authors · 2026-05-20
> **Response to Reviewer 3LEy**
>
> We thank the reviewer for the thoughtful and constructive review! We are encouraged that reviewer find our work has clever pipeline and is well-motivated. Below we address each of the requested changes and outline the specific modifcations we will make:
>
> ## 1. Defining K and P in Section 2.2
> We agree that the $(K \times P)$ parameter block needs explicit definitions. We will add a sentence stating that $K$ denotes the number of Gaussians (defined later as $K = 10,144$ in our experiments) and $P = 59$ denotes the number of parameters per Gaussian, broken down as: $3$ (position) + $3$ (scale) + $4$ (rotation quaternion) + $1$ (opacity) + $48$ (spherical harmonic coefficients). This will be introduced at the point of first use.
>
> ## 2. Undefined symbols in Section 3
> We will add explicit definitions for all variables in the alpha-blending equation. Specifically, we will clarify that $d$ indexes the Gaussians intersected along a camera ray sorted by depth, $c_d$ is the evaluated color from spherical harmonic coefficients for Gaussian $d$, and $\alpha_d$ is the effective opacity incorporating the Gaussian's learned opacity and its projected 2D density at the pixel location. We will also improve notation clarity in Sections 3.3.1 and 3.5: in Section 3.3.1, we will explicitly define $w \in R^{|w|}$ as the pose-invariant identity code (with $|w| = 512$), $e_k \in R^{|e|}$ as the per-Gaussian embedding (with $|e| = 32$), and each component of the residual $\Delta G_k = \{\Delta \mu_k, \Delta s_k,  \Delta q_k,  \Delta \alpha_k, \Delta c_k\}$ with its dimensionality ($R^3, R^3, R^4, R^1, R^{48}$ respectively). In Section 3.5, we will define $w_{k,j}$ as the fixed linear blend skinning weight of Gaussian $k$ with respect to FLAME joint $j$, $A_j$ as the joint transformation matrix, and $T_k$ as the blended transformation for Gaussian $k$. We will also define the loss weighting terms $\lambda_1$ through $\lambda_5$ in Eq. (1) and $λ_{cos}$ in Eq. (2) with their numerical values (currently provided in the appendix).
>
> ##  3. Clarifying "canonical FLAME coordinates" for positional encodings in Section 3.3.1
> We will add a precise definition of this term. FLAME [1] is a parametric 3D head model that represents facial geometry as a triangular mesh with fixed topology ($9,976$ faces). The "canonical FLAME coordinates" are simply the 3D (x, y, z) positions of these triangle centers when the mesh is in its rest state (neutral expression, zero pose). Because the mesh topology is fixed, the k-th triangle always corresponds to the same facial region across all subjects. We apply sinusoidal positional encodings to these coordinates to obtain the per-Gaussian embeddings $e_k$.
>
> ## 4. Clarifying multi-view training vs. single-view inference
> We agree this is an important point. We will add an explicit clarification early in the introduction (and reiterate in Section 3) stating that while our training pipeline leverages multi-view data to construct the canonical template and train the encoder-decoder network, the trained system requires only a single image at inference time. The multi-view data is used solely to build the geometric prior offline; no multi-view capture is needed for new subjects.
>
> We plan to incorporate all of these changes into the revised manuscript once all the reviews are available, so that we can address all feedback in a single comprehensive revision. Thank you again for your time and helpful suggestions.
>
> [1] Li et al. Learning a model of facial shape and expression from 4D scans.

---

### Review · Reviewer_dAL6 · 2026-06-05

**Summary Of Contributions:**

This paper proposes FastAvatar, a novel framework designed to generate high-fidelity 3DGS face avatars from a single unconstrained image rapidly. To tackle the highly ill-posed challenge of inferring hundreds of thousands of 3DGS parameters from a single view, the authors draw inspiration from classical 3DMMs, constructing a canonical 3DGS template anchored to the FLAME mesh topology, which is obtained by averaging optimized multi-view subject data.
Building upon this geometric prior, the paper introduces an efficient two-stage reconstruction pipeline. First, a feed-forward encoder-decoder network trained via a decoupled strategy predicts local Gaussian residuals with respect to the template in milliseconds. Second, a lightweight inference-time refinement is applied.
Extensive evaluations on the NeRSemble benchmark demonstrate that FastAvatar achieves state-of-the-art single-view reconstruction accuracy. Owing to the inherent parametric mesh binding, the reconstructed 3DGS models naturally support real-time expression animation and reenactment via LBS.

**Audience:**

Yes

**Audience Explanation:**

Yes. 3DGS is currently one of the most active and rapidly evolving topics in the machine learning community (especially computer vision). Researchers working on efficient novel-view synthesis will be highly interested in the paper's approach to bypassing the notoriously lengthy per-subject optimization typically required by NeRF and standard 3DGS frameworks.

**Claims And Evidence:**

Yes

**Claims Explanation:**

Yes, the core claims made in the submission are generally supported by accurate, convincing, and clear empirical evidence, although there are a few methodological caveats that should be addressed in a peer review.

**Requested Changes:**

Critical:
- The current experimental validation relies almost entirely on the NeRSemble dataset. While the authors successfully hold out specific identities for testing, NeRSemble is captured in a highly controlled laboratory environment with uniform lighting, clean backgrounds, and specific camera intrinsics. To fully substantiate the claim that FastAvatar can generate robust 3D models from a single arbitrarily posed image, the readers need to see how the method performs on unconstrained, in-the-wild data. So I request the authors include qualitative results on standard in-the-wild datasets such as FFHQ, VFHQ, or CelebA-HQ.
- The paper claims to support real-time animation via FLAME-guided LBS. However, it is a well-known issue that 3D Gaussian Splatting is highly sensitive to non-rigid deformations; large LBS-driven movements often lead to severe stretching, tearing, or splitting artifacts in the rendered Gaussians. The current static images are insufficient to evaluate the temporal stability and visual quality of the avatar under dynamic motion.

Not Critical:
- The single-image inference pipeline relies on extracting accurate FLAME parameters via external trackers (like EMOCA) to align the test image. The paper does not provide ablation evidence demonstrating how robust FastAvatar is if this external pose estimation step fails, produces noisy data, or encounters extreme facial expressions.
- The current title "FastAvatar" is quite generic and potentially overlaps with existing literature. More importantly, it fails to highlight the unique technical contribution of the work: the construction of a fully animatable avatar system that relies exclusively on a 3DGS + LBS paradigm, effectively bypassing the need for explicit mesh rendering proxies. A more descriptive title could better attract the target research community interested in mesh-free neural representations.
- Please ensure that all citations in the final manuscript adhere to the required LaTeX format, ensuring consistent use of \citet{} for textual citations and \citep{} for parenthetical citations.

---

> ### Author Response · Authors · 2026-06-17
> **Response to Reviewer dAL6**
>
> We thank the reviewer for the positive and constructive review, and for recognizing the relevance of our work to the 3DGS and novel-view synthesis community. We appreciate your careful reading and the actionable suggestions. Below we outline how we will address each point.
>
> ## Critical 1: Qualitative results on in-the-wild data (FFHQ, VFHQ, CelebA-HQ)
>
> We agree that demonstrating performance beyond the controlled NeRSemble setting is important for substantiating our single-image robustness claim. The main challenge of using in-the-wild single images as input is the fact that they lack ground truth multi-views. Nevertheless, we will add qualitative results on in-the-wild images, using samples from FFHQ and CelebA-HQ.
>
> ## Critical 2: Temporal stability of LBS-driven animation
>
> This is a fair point, and we agree that static frames cannot fully convey the visual quality of the avatar under motion. We will include example videos in the supplementary material showing LBS-driven animation, which demonstrate that the avatar remains stable under mild motion but exhibits artifacts under larger expression changes. We will also add a discussion of this behavior in the main text, characterizing the range of motion over which the avatar remains stable and the failure modes that arise beyond it.
>
> ## Not Critical 1: Robustness to FLAME tracker errors
> This is indeed an interesting point. According to our knowledge, FLAME-based trackers like emoca, deca are mostly stable but a few outliers: extreme side/profile views, extreme expressions, occlusions. We have identified some cases of such inputs from CelebA, and we will add a visualization + discussion on that, to further stress test our framewowrk and its limit.
>
> ## Not Critical 2: Title
> We appreciate the suggestion. We will reconsider the title to better reflect the technical contribution, for example, explicitly stating the template + residual framework, and reduce potential overlap with existing literature, while keeping it accessible to the target community.
>
> ## Not Critical 3: Citation formatting
> We will perform a consistency pass to ensure correct use of \citet{} for textual citations and \citep{} for parenthetical citations throughout the manuscript.
>
> We plan to incorporate all of these changes into the revised manuscript once all the reviews are available, so that we can address all feedback in a single comprehensive revision. Thank you again for your time and helpful suggestions.

---

> ### Author Response · Authors · 2026-06-29
>
> We thank the reviewer again for the positive and constructive review! As detailed in our earlier response, we have incorporated the promised changes into the revised manuscript; they are highlighted in blue. For quick reference, here is where each appears:
>
> | Concern | Change | Location |
> |---|---|---|
> | Critical 1 — Qualitative results on in-the-wild data (FFHQ / CelebA-HQ) | Reconstructions from real in-the-wild images; we first run a face detector to crop and mask the input, then run FastAvatar | Sec. 4.4, Fig. 8, Fig. 11 |
> | Critical 2 — Temporal stability of LBS-driven animation | Added a free-form LBS-driven animation sequence and a discussion of the stable range of motion vs. the artifacts under large deformations; a sample animation video is included in the supplementary zip | Sec. 4.3, Fig. 7; sample video in Supplementary |
> | Not Critical 1 — Robustness to FLAME tracker errors | Stress test over extreme/profile views, occlusions, and expressions that challenge the tracker, with discussion; when the tracker fails we omit or reduce the test-time refinement | Sec. 4.7, Fig. 11 |
> | Not Critical 3 — Citation formatting | Consistency pass using `\citet{}` for textual and `\citep{}` for parenthetical citations | Throughout |
>
> For all of these revisions, please refer to the Summary of Changes above and to the revised submission (changes highlighted in blue).

---

### Review · Reviewer_mi7D · 2026-06-21

**Summary Of Contributions:**

This paper introduces FastAvatar, a hybrid framework that generates animatable 3D Gaussian Splatting (3DGS) face avatars from a single image using a novel "template-plus-residuals" pipeline, which predicts structural and appearance deviations from a learned canonical template. Its primary strength lies in a significant optimization speedup, achieving coarse geometry in under 10ms and photorealistic refinement in 3 seconds, while maintaining state-of-the-art visual metrics and seamlessly supporting downstream expression animation.

**Audience:**

Yes

**Audience Explanation:**

3D Gaussian Splatting, neural rendering, and implicit 3D representations are among the most active and rapidly growing subfields within machine learning community. Specifically, the task of generating high-fidelity, animatable digital humans from extremely limited data (a single image) remains a highly complex challenge. The finding that a hybrid "template-plus-residuals" framework can successfully bypass heavy optimization loops without sacrificing structural or visual fidelity is of high interest for the community.

**Claims And Evidence:**

Yes

**Claims Explanation:**

The authors provide an extensive quantitative and qualitative evidence to back up their primary claims of speed, quality, and view-invariance. Evaluation on standard datasets like Nersemble demonstrates state-of-the-art performance against comparable single-view baselines across standard rendering metrics (PSNR, SSIM, LPIPS). The claimed speedup is explicitly demonstrated via various runtime evaluations. Extensive ablation studies successfully validate the necessity of both the decoupled identity embedding and the hybrid two-stage generation process.

**Requested Changes:**

* Since the framework operates on a single input image, generating a full 3D head avatar requires synthesis for occluded regions. The authors could provide a qualitative or quantitative evaluation tracking identity consistency across extreme yaw rotations. Specifically, evaluate if the model suffers from identity asymmetry or when the occluded side defaults too heavily to a generic canonical face.

* The authors provide great analysis in supplementary material regarding in-the-wild generalization and failure modes for out-of-distribution geometries (e.g., extreme hairstyles, glasses). It would significantly strengthen the main paper if a representative visual example of these failure cases and in-the-wild performance were moved to the main text's limitation and results sections.

* The 3-second test-time optimization stage updates parameters directly against the single input image to maximize photorealism. However, real-world images often suffer from sensor noise, low resolution, or harsh directional shadows. It would significantly strengthen the paper to include a brief robustness analysis of the test-time refinement phase.

* Single-view 3DGS risks baking directional lighting into Gaussian colors, causing illumination to unrealistically rotate with the head during animation. That would be great to evaluate how the framework handles extremal / harsh lighting.

---

> ### Author Response · Authors · 2026-06-29
>
> We thank the reviewer for the constructive and detailed feedback! We have revised the manuscript to address each of the requested changes; all changes are highlighted in blue. For quick reference, here is what we changed and where it appears:
>
> | Concern | Change | Location |
> |---|---|---|
> | Identity consistency across extreme yaw rotations (identity asymmetry / occluded side) | Added a stress test that discusses identity asymmetry on the hallucinated occluded side under extreme yaw, and a table reporting how PSNR, SSIM, and identity similarity (ArcFace cosine) degrade with the total angular deviation of the input from the frontal view, directly quantifying identity consistency under increasing yaw | Sec. 4.7, Fig. 11, Table 3 |
> | Move representative failure cases and in-the-wild performance from the supplementary to the main results / limitations | Added in-the-wild results and a stress-test / failure-case subsection in the main results, and updated the Limitations to point to these failure cases | Sec. 4.4 (Fig. 8), Sec. 4.7 (Fig. 11), Sec. 5 |
> | Robustness of the test-time refinement to real-world inputs (sensor noise, low resolution, harsh shadows) | Stress test includes low-resolution and non-ideal-lighting inputs; we discuss when the single-image refinement amplifies input defects, and note that we omit or reduce it when the FLAME tracker fails | Sec. 4.7, Fig. 11 |
> | Handling of extremal / harsh lighting (illumination baked into Gaussian colors) | Generated inputs under varied lighting via single-image portrait relighting and show that FastAvatar bakes lighting into the appearance parameters, so novel views carry the input lighting (not always precisely) | Sec. 4.7, Fig. 12 |
>
> For all of these revisions, please refer to the Summary of Changes above and to the revised submission (changes highlighted in blue).